# Pre-stimulus beta power mediates explicit and implicit perceptual biases in distinct cortical areas
Carina Forster [1,2] ✉, Tilman Stephani [1,7], Martin Grund [1], Eleni Panagoulas[1,3,4], Esra Al[1,5], Simon M. Hofmann[1,6], Vadim V. Nikulin[1] & Arno Villringer [1,3,4]

Perception is biased by expectations and previous actions. Pre-stimulus brain oscillations are a potential candidate for implementing biases in the brain. In two EEG studies (43 and 39 participants) on somatosensory near-threshold detection, we investigated the pre-stimulus neural correlates of an (implicit) previous choice bias and an explicit bias. The explicit bias was introduced by informing participants about stimulus probability on a single-trial level (volatile context) or block-wise (stable context). Behavioural analysis confirmed adjustments in the decision criterion and confidence ratings according to the cued probabilities and previous choice-induced biases. Pre-stimulus beta power with distinct sources in sensory and higher-order cortical areas predicted explicit and implicit biases, respectively, on a single subject level and partially mediated the impact of previous choice and stimulus probability on the detection response. We suggest pre-stimulus beta oscillations in distinct brain areas as a neural correlate of explicit and implicit biases in somatosensory perception.

Perception is biased by our knowledge of what is probable. This has been successfully formalised in the Bayesian brain theory[1], which assumes an integration of prior beliefs and the sensory signal. According to this theory, the brain relies on prior beliefs when sensory signals are "uncertain", i.e. weak. Indeed, providing human observers with information about stimulus probability has been shown to bias perceptual reports of weak visual stimuli[2]. In such situations, participants adjust their decision criterion while sensitivity to the sensory signal remains unaffected[3].

Additionally, providing individuals with prior information about the probability of a stimulus has been shown to impact confidence in a perceptual decision[4]. Besides external and explicit information about the probability of upcoming stimuli, the perceptual choice history also influences decision-making, with stronger choice-history biases in ambiguous perceptual decisions[5]. In the following, we define the bias induced by choice history as implicit bias, as it arises from internal processes that the experimenter does not explicitly control. It is important to note that the terms implicit and explicit, as used throughout this manuscript, do not pertain to participants' level of awareness. Rather, we classify the choice-history bias as implicit because it emerges in the absence of explicit information provided

to the observer. Although studies have demonstrated the adaptability of choice-history biases to participants' models of the environment[6,7], whether and how they interact with explicit biases has not been investigated.

While a substantial body of research has established that explicit expectations and choice history shape perception at the behavioural level, a notable gap remains in our understanding of the distinct or shared neural processes[2,8–10]. Gustatory stimulation in rats in combination with computational modelling showed that stimulus expectations modulated pre-stimulus metastable dynamics of neural activity[11]. Using MEG recordings in humans, Kok and colleagues[12] reported that neural representations of expected visual stimuli in pre-stimulus activity closely resembled the evoked activity following the onset of expected stimuli, suggesting that the brain generates stimulus templates in the sensory cortex to anticipate expected inputs. Pre-stimulus activity has also been shown to reflect choice-history biases[13,14]. Surprisingly, there have been no studies so far that investigated the effect of implicit and explicit biases on pre-stimulus activity in the same paradigm.

EEG and MEG studies have emphasised the impact of pre-stimulus oscillations, particularly in the alpha and beta frequency range, on

[1]Max Planck Institute for Human Cognitive and Brain Sciences, Department of Neurology, Leipzig, Germany. [2]Charité – Universitätsmedizin Berlin, Bernstein Center for Computational Neuroscience, Berlin, Germany. [3]Charité – Universitätsmedizin Berlin, corporate member of Freie Universität Berlin and Humboldt-Universität zu Berlin, BCAN Berlin Centre for Advanced Neuroimaging, Berlin, Germany. [4]Humboldt - Universität zu Berlin, Berlin School of Mind and Brain, Berlin, Germany. [5]Department of Psychiatry, Columbia University, New York, NY, USA. [6]Max Planck Institute for Human Cognitive and Brain Sciences, Neural Data Science and Statistical Computing Group, Leipzig, Germany. [7]Present address: Donders Institute for Brain, Cognition and Behaviour, Radboud University, Nijmegen, The Netherlands. ✉e-mail: carinaforster0611@gmail.com

perceptual decision-making. Multiple studies have shown that pre-stimulus alpha/beta power in the somatosensory cortex negatively correlates with detection rates for a subsequent weak tactile stimulus[15–20]. According to the baseline sensory excitability model (BSEM)[21], pre-stimulus alpha power correlates positively with the participants' decision criterion in detection tasks. Indeed, recent studies in the visual[22] and the somatosensory domain[23], confirmed a positive correlation between spontaneous modulations of pre-stimulus alpha power and the participants' criterion. So far, only two studies have experimentally manipulated the decision criterion in humans. Both studies employed a visual detection task, inducing criterion changes through reward strategy[24] or priming[25]. Kloosterman et al.[24] incentivised either a liberal or conservative criterion and showed a correlation between visual pre-stimulus alpha power and the change in criterion. Zhou et al.[25], however, did not find a significant correlation between visual pre-stimulus alpha power and criterion changes that were induced via priming. While pre-stimulus alpha power has been suggested as a neural correlate of criterion changes, two experimental studies did not provide consistent evidence supporting the role of alpha power.

Top-down predictions have also been related to modulations of beta power[26,27]. Consistent with this notion, the control of goal-directed sensory processing has been linked to beta power and beta bursts[28]. Beta power has also been proposed to be related to the maintenance of cognitive states[29], with evidence pointing to beta power not only maintaining but also reactivating cognitive states that are required for the current task[30], which again supports the idea of a pre-stimulus template of expectations as has been suggested by ref. 12.

In summary, previous research has proposed that pre-stimulus alpha and beta power serve as neural correlates of stimulus expectations. However, there remains no consensus on the specific roles of pre-stimulus alpha and beta power in shaping decision biases and whether they are linked to explicit and implicit biases.

To address this gap, we conducted two complementary EEG studies that differed in the temporal context of stimulus occurrence[31]. We hypothesised that human observers would use information about stimulus probability to adjust their detection and confidence ratings and exhibit choice-history biases. Based on previous findings, these biases should be reflected in pre-stimulus alpha and/or beta frequency oscillations in sensory areas.

## Methods
### Participants
Forty-three healthy, young volunteers (22 women, 21 men, age: 26.7 ± 4.4 years, [mean ± SD], range: 21 to 35 years) were recruited for the first study and forty for the second study (23 women, 17 men, age: 25.7 ± 3.9 years, range: 19 to 35 years) from the database of the Max Planck Institute for Human Cognitive and Brain Sciences, Leipzig, Germany. Gender was determined via participants' self-reports. We did not collect data on race or ethnicity. All participants reported to be right-handed. Both studies were approved by the Ethics Committee of the University of Leipzig's Medical Faculty (462/15-ek). All participants provided written informed consent and were reimbursed 9.00 Euro per hour for their participation in the first study and, due to a change of regulations, 12.00 Euro per hour in the second study. The study was not preregistered.

### Experimental setup
This study aimed to investigate how stimulus probability influences tactile perception and confidence in humans. All data were acquired at the Max Planck Institute for Human Cognitive and Brain Sciences, Leipzig. Electrical finger nerve stimulation was performed with a constant-current stimulator (DS5 Isolated Bipolar Current Stimulator (RRID:SCR_018001) using single square-wave pulses with a duration of 200 µs. A waveform generator NI USB-6343 (National Instruments) and custom MATLAB scripts using the Data Acquisition Toolbox (MATLAB (RRID:SCR_001622)) were used to control the stimulation device. Steel wire ring electrodes were placed on the middle (anode) and the proximal (cathode) phalanx of the index finger on

the left hand. The experimental procedure was controlled by custom MATLAB scripts using the Psychophysics Toolbox (RRID:SCR_002881).

### Experimental paradigm
At the beginning of the experimental session, we recorded 5 min of resting state EEG with eyes open while participants were seated in a comfortable chair in the EEG cabin and fixated on a grey cross on the screen. Next, participants were familiarised with the electrical finger nerve stimulation. An automatic threshold assessment was performed to determine the stimulus intensity corresponding to the somatosensory detection threshold. The threshold assessment entailed an up-and-down procedure (40 trials in the first run and 25 trials in subsequent runs) which served as a prior for the following Bayesian approach (psi method; 45 trials in the first run and 25 trials in subsequent runs) from the Palamedes Toolbox[32] (RRID:SCR_006521) and finished with a test block (five trials without stimulation and ten trials with stimulation intensity at the threshold estimate by psi method). Based on the test block results for the psi method threshold estimate and weighting in the results of the up-and-down procedure, the experimenter selected a near-threshold intensity, approximately at a 60% detection rate.

### Stable environment
In each trial, participants were instructed to indicate whether they perceived a weak somatosensory stimulus (Yes/No) and then to state their retrospective confidence in this decision using a binary rating (Confident/Unconfident). Participants were informed that they would receive cues about the probability of a stimulus on a given trial throughout the experiment. In study 1, a cue appeared at the beginning of each condition block for three seconds and was valid until the next probability cue appeared (12 trials, the number of trials in a block was not explicitly mentioned). The high stimulus probability condition (75%) instructed participants that there was a 75% chance of a stimulus on a given trial, whereas the low probability condition (25%) informed participants that there was a 25% chance of a stimulus on a given trial. Crucially, those cues were always valid, and participants were explicitly told about those contingencies. To ensure motivation throughout the experiment, we provided participants with feedback (percentage correct displayed for 2.5 s) at the end of each probability block. Each of the five blocks contained 144 trials with six repetitions for each probability condition. Thus, half of the trials included a near-threshold electrical pulse (mean intensity = 1.9 mA, range: 0.9–3.7 mA). The stimulus intensity was adjusted after each experimental block if the detection rate was no longer near-threshold. The other half of the trials did not contain a stimulus (noise trials). The order of signal and catch trials and probability condition miniblocks was pseudo-randomised for each experimental block and participant. Participants responded with the right index finger on a two-button box. Before starting the main experiment, participants completed a training block of 48 trials.

### Volatile environment
In the second study, participants were instructed to indicate whether they perceived a weak somatosensory stimulus (Yes/No) and then to state their retrospective confidence in this decision using a binary rating (Confident/Unconfident) in each trial. Participants were informed that they would receive cues about the probability of a stimulus before each trial throughout the experiment. In the second study, either an orange or blue circle appeared at the beginning of each trial for 1 s and indicated either high or low stimulus probability (the colour assignment was randomised across participants). The high stimulus probability condition (75%) instructed participants that there was a 75% chance of a stimulus on a given trial, whereas the low probability condition (25%) informed participants that there was a 25% chance of a stimulus on a given trial. Crucially, these cues were valid, and participants were explicitly told about these contingencies. To motivate participants throughout the experiment, we provided participants with feedback (percentage correct displayed for 2.5 s) after 30 trials. Each of the five blocks contained 120 trials with an equal amount of high and low

stimulus probability trials overall. Thus, half of the trials contained a near-threshold electrical pulse (mean intensity = 2.1 mA, range: 1.1–3.6 mA). The other half of the trials did not contain a stimulus (catch trials). The order of high and low probability trials and stimulus and noise trials was pseudo-randomised within blocks using the Shuffle function in Matlab. Participants responded with the right index finger on a two-button box. Before starting the main experiment, participants completed a training block of 40 trials. Additionally, we asked participants about the colour-probability mapping after each experimental block to ensure the probability cues were correctly remembered.

### Behavioural data preprocessing
**Stable environment (study 1).** Across all participants, we collected 30.816 trials from 43 participants. Thirty-five trials were rejected due to missing EEG triggers (EEG recording started too late). Next, we removed 39 no-response trials (no button pressed within 2.5 s) and 45 trials with a detection response time of less than 100 ms. We excluded eleven blocks that had a hit rate greater than 90% or less than 20% and resulted from suboptimal threshold estimation. These criteria were determined based on values that deviated more than three times the standard deviation from the mean hit rate calculated across all blocks and participants (mean hit rate: 58%, standard deviation: 10%). Additionally, for one participant, one block was excluded based on a false alarm rate of more than 40%. After behavioural preprocessing, 28.977 trials remained.

**Volatile environment (study 2).** We collected 24.000 trials from 40 participants. Data from one participant was rejected due to technical issues with the somatosensory stimulation device. From the remaining 39 participants, 12 trials were rejected due to missing EEG triggers (the battery died or EEG recording started too late). Next, we removed 133 no-response trials (no button pressed within 2.5 s) and 102 trials with a detection response time of less than 100 ms. We excluded ten blocks that had a hit rate greater than 90% or less than 20%. Additionally, two blocks were excluded based on a false alarm rate of more than 40%. One block was excluded because the false alarm rate was higher than the hit rate. We used the same exclusion criteria as in the first study (mean hit rate: 53%, standard deviation: 12%). After behavioural data cleaning, 21.593 trials from 39 participants remained.

### Signal detection theory analysis
We employed signal detection theory (SDT)[33] to examine the sensitivity (Dprime) and response bias (criterion c) in this study. SDT allows us to distinguish between the ability to detect a signal (e.g. the presence of a stimulus) from a general tendency to report either stimulus presence or absence.

**Sensitivity (Dprime).** Sensitivity, also known as Dprime, quantifies the ability to discriminate between signal and noise. It is calculated using the following formula:

$$d' = z(hit\ rate) - z(false\ alarm\ rate) \qquad (1)$$

where z(hit rate) represents the z-score of the hit rate (proportion of correct responses when the signal is present), and z(false alarm rate) represents the z-score of the false alarm rate (proportion of incorrect responses when the signal is absent).

**Response bias (criterion c).** Response bias, represented by criterion c, assesses the individual's tendency to respond "yes" or "no" irrespective of the presence (or absence) of the stimulus. It is calculated using the formula:

$$c = -0.5 * [z(hit\ rate) + z(false\ alarm\ rate)] \qquad (2)$$

In our analysis, we computed d' and c for each participant to examine their sensitivity to the stimuli and response bias. Higher values of d' indicate

better sensitivity, while a positive value of c reflects a bias towards responding "no" (conservative criterion). The Hautus log-linear correction method[34] was employed to address zero false alarm rates.

### Statistical analysis: behaviour
Nonparametric Wilcoxon signed-rank tests were applied to compare the paired samples in both studies. Tests for normality indicated non-normal distributions of criterion and sensitivity in the first study. The Wilcoxon signed-rank test is robust to deviations from normality and assesses whether the median difference between paired observations is significantly different from zero. All tests are two-sided if not stated otherwise in the results section.

**Effect size calculation.** We calculated Cohen's d as a measure of effect size for the main behavioural results. Cohen's d is a measure of standardised mean difference, commonly used to assess the magnitude of effect in paired data. For each comparison, Cohen's *d* was computed using the formula:

$$Cohen's\ d = mean\ of\ paired\ differences / \\ standard\ deviation\ of\ paired\ differences \qquad (3)$$

Following Cohen's guidelines, effect sizes were interpreted as small (*d* = 0.2), medium (*d* = 0.5), or large (*d* = 0.8).

### Estimation of signal detection theory parameters using generalised linear mixed-effects models
To investigate the influence of single-trial variables on signal detection theory (SDT) parameters[35], we employed generalised linear mixed-effects models (GLMMs) using the lme4 package[36] in R (version 4.3.3). We specified a GLMM with a binomial distribution and probit link function to account for the binary nature of the detection or confidence response (signal present or absent response, high or low confidence). In the simplest model, the intercept represents the overall criterion (c), and the regressor that codes whether the trial contained a stimulus corresponds to Dprime[37] (regression formula based on lme4 syntax: detection_response ~ stimulus + (stimulus| subject)). Random intercepts were included in all models, random slopes for the main effects and interaction effects only if the models did not show warnings about singularity[38]. To assess collinearity among the predictor variables in our analysis, we utilised the "check_collinearity" function from the 'performance' package and to ensure convergence of the models the check_converge function[39] in R. Variance inflation factor (VIF) values were computed for all models and parameters, demonstrating low values (VIF <3) and indicating the absence of substantial multicollinearity for all SDT GLMMs. We used the emmeans (estimated marginal means) R package[40] for FDR-corrected post hoc tests on our fitted GLMMs. Model summaries are based on the modelsummary package[41], and interaction effects plots are based on the plot_model() function provided by the sjPlot package[42]. We compared Akaike Information Criterion (AIC) measures of each model and considered a difference greater than 10 as a meaningful difference in model fit[43]. As an independent source of model fit, we compared models using the anova() function from the stats package (v. 3.6.2) in R, which compares nested models based on likelihood ratio tests.

### EEG recordings
EEG was recorded from 62 scalp positions distributed over both hemispheres according to the international 10–10 system, using a commercial EEG acquisition system (Standard 64ch actiCap Snap, BrainAmp; Brain Products, Brain Products (RRID:SCR_009443). The mid-frontal electrode (FCz) was used as the reference, and a mid-frontal electrode was placed on the middle part of the forehead (between FP1 and FP2) as ground. One additional electrode was used to measure electroocular activity (placed below the right eye) and ECG activity (placed below the left clavicle), respectively. Electrode impedance was kept at ≤10 kΩ for all channels. EEG was recorded with an online bandpass filter from 0.015 Hz to 1 kHz and

digitised with a sampling rate of 2.5 kHz for study 1 (stable environment) and 1 kHz for study 2 (volatile environment).

## EEG - preprocessing

We used custom Python scripts and the MNE Python package (MNE software (RRID:SCR_005972)[44] to analyse the EEG data of both datasets. We applied a bandpass filter between 0.1 and 40 Hz to the raw EEG data with the following IIR filter parameters: Butterworth zero-phase (two-pass forward and reverse) non-causal filter, filter order 16 (effective, after forward-backwards, cutoffs at 0.10, 40.00 Hz: −6.02, −6.02 dB). Next, we linearly detrended and epoched the data between −1 s before stimulation onset and 1 s after stimulation onset. After downsampling the data to 250 Hz, we used the RANSAC package[45] to detect bad channels (channels interpolated study 1: mean = 1.0, max. = 4, study 2: mean = 0.7, max. = 5). Next, we ran ICA on the 1 Hz high-pass filtered, epoched data using the extended infomax algorithm[46]. We correlated ICA components with EOG and ECG activity and rejected components that correlated strongly with eye movements and cardiac artefacts from the 0.1 Hz filtered data (components rejected in study 1: mean = 3.5, max. = 7, study 2: mean = 3.5, max. = 6). Finally, we re-referenced the data to the common average of all EEG channels.

## Evoked potentials

Electrode CP4 was selected as the channel of interest (COI) based on the post-stimulus contrast between signal and noise trials. To compare the experimental conditions, we initially averaged the epochs of the defined condition for each participant and ensured that the epoch counts were equalised across all conditions. Subsequently, we calculated the average evoked data across all participants. We used a baseline window of 100 ms before stimulation onset and subtracted the averaged activity from the post-stimulus activity for each epoch. To determine the channel of interest, we compared signal and noise trials in the post-stimulus window and focused on the earliest somatosensory evoked potential, which was in our paradigm and with our data preprocessing a positive evoked potential around 50 ms post-stimulation (which we refer to as P50 component in the following).

## Time-frequency representation in the pre-stimulus window

We used the multitaper method[47] to compute the time-frequency representation in the frequency range of 3 to 34 Hz with a resolution of 1 Hz and a time-bandwidth of 4 Hz. The number of cycles for each frequency was defined as a frequency divided by four cycles (resulting in a temporal resolution of 125 ms). We mirrored the data on both ends to minimise edge artefacts. The number of epochs was matched between conditions using the minimum time difference method implemented in MNE Python. For the contrast between high and low stimulus probability in the stable environment, we calculated averaged time-frequency representations separately for previous hits (previous yes response in previous signal trial), misses (previous no response in previous signal trial) and correct rejections (previous no response in previous noise trial) for the high and low stimulus probability conditions. Next, we subtracted the respective contrasts, e.g., high previous hits vs. low previous hits. Finally, we averaged the time-frequency representations of high vs. low stimulus probability corrected for previous trial history. The design in the volatile environment allowed us to average over high and low probability trials without controlling for previous trial characteristics, as the previous trial history was randomised (Suppl. Fig. 1.4). To specifically examine the influence of past choices on power modulations, we restricted our analysis to trials occurring under high-probability conditions and those following a signal in the stable environment. In the volatile environment, our analysis targeted trials with identical probability cues in both the preceding and current trials in the high-probability condition.

## Cluster-based permutation tests

For statistical comparisons of the time and frequency domain, we used threshold-free cluster-based permutation testing implemented in MNE with a cluster and statistical threshold of $p = 0.05$ and 10.000 permutations.

The test statistic was a two-tailed one-sample $t$-test, as we tested whether the difference between conditions was significantly different from zero. Adjacency was defined over time (neighbouring time points are considered as adjacent) and frequency (neighbouring frequencies are considered as adjacent). We used a step size of 0.1 and an initial t-threshold of 0 for threshold-free cluster enhancement[48]. The MNE defaults for the cluster height (2) and cluster extent (0.5) were used.

## DICS source localisation of pre-stimulus power

Source localisation was performed using dynamic imaging of coherent sources (DICS)[49] beamforming implemented in MNE. The source space encompassed a total of 8196 individual sources (5 mm spacing between sources). We used the MNE standard BEM (5120-5120-5120) model based on the freesurfer average brain and standard electrode positions. For each participant, source localisation was computed for the beta band frequency range where we observed the strongest modulations in sensor space (15–25 Hz) in the pre-stimulus window, averaged over 600 ms before stimulation onset (700 to 100 ms before stimulation onset). Cross-spectral density was computed for all epochs and separately for the high probability and low probability epochs, and the previous yes and no previous no responses. To compute source power, we estimated cross-spectral density for each epoch using Morlet wavelets with the cycles linearly increasing per frequency (frequency/4 cycles). Next, we averaged over all frequencies and computed a common spatial filter, before we applied the filter to the data of each condition. The filter was computed for the orientation that maximised power. Finally, we subtracted the source power of low probability trials from the source power of high-probability trials before we averaged the contrast over participants. To visualise the contrast, we ran a permutation t-test (10 000 permutations) for all sources, and we highlight the source with the highest $t$-values on the fsaverage template brain. We validated our beamformer on the beta power desynchronisation for the post-stimulus contrast between signal and noise trials, which showed—as expected—a source in the postcentral gyrus (Suppl. Fig. 2.4).

## Pre-stimulus source power per trial

We created a mask based on the contrast for stimulus probability or previous response to extract pre-stimulus power in source space. Voxels with the highest $t$-values for each contrast were defined, and power for each trial was averaged over those vertices. For the brain-behaviour modelling, the source power was log-transformed, z-scored and detrended over all trials and for each experimental block. Trials with z-scored source power >3 or <−3 were excluded from the analysis. Pre-stimulus power was entered as a continuous regressor in our models and only binned for visualisation.

## Mediation analysis

We conducted mediation analysis using the "mediation" package[50] in R. The mediation analysis examined the mediating role of pre-stimulus power on the relationship between stimulus probability (independent variable) and the detection response (dependent variable). To conduct the mediation analysis, we set up a linear mixed-effects model with the mediator as the dependent variable (pre-stimulus power) and the independent variable as the predictor (stimulus probability). Next, we fitted a GLMM with the detection response as the outcome variable (dependent variable) and the mediator (beta power) and the independent variables as predictors. We included the stimulus on each trial as a covariate in our model. The same mediation model was fit with the previous choice as an independent variable. Finally, we computed the indirect effect, the total effect and the direct effect. The 95% confidence intervals of the regression weights were estimated using quasi-Bayesian approximation with 1000 Monte-Carlo permutations.

## Reporting summary

Further information on research design is available in the Nature Portfolio Reporting Summary linked to this article.

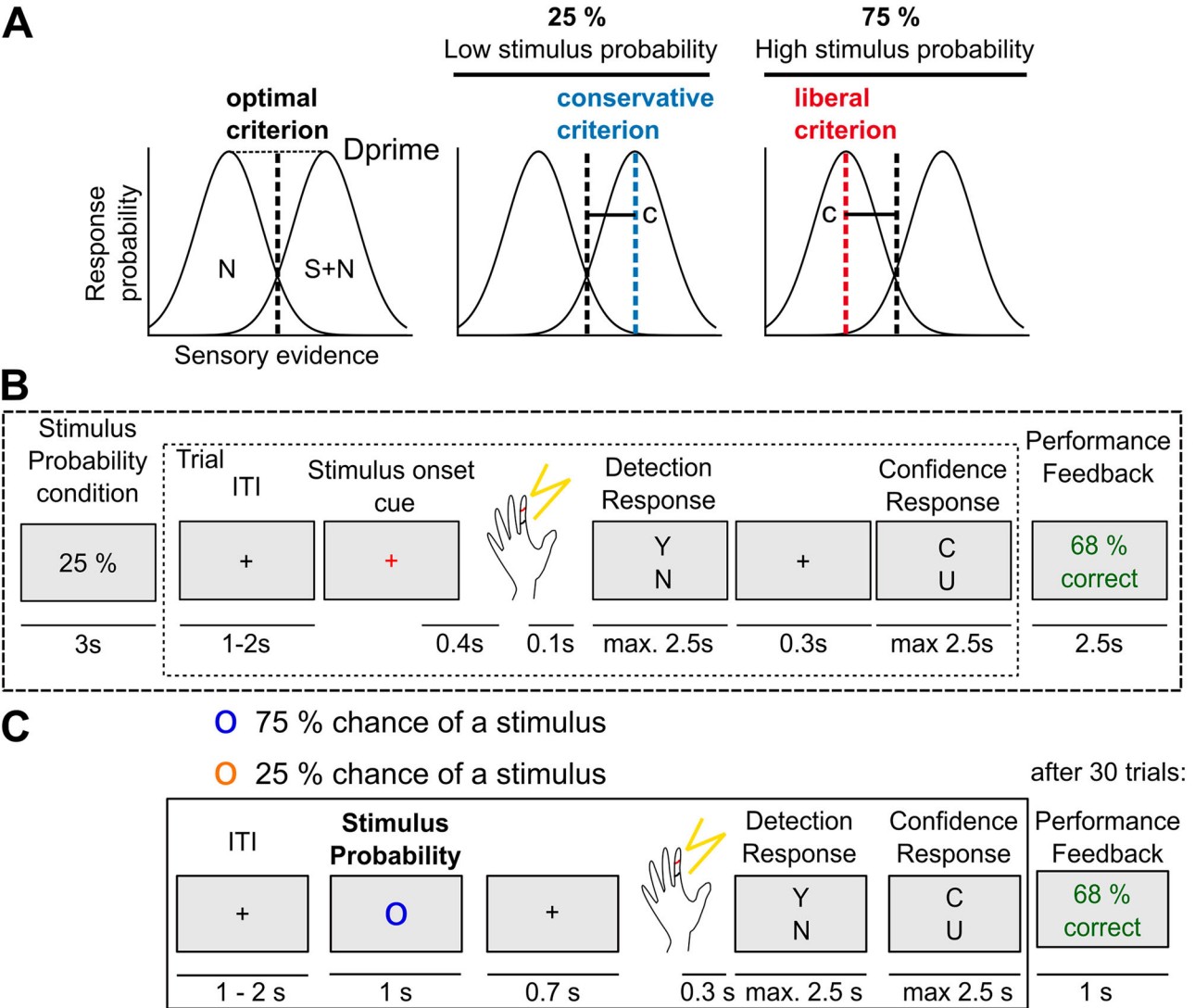

**Fig. 1 | Stimulus probability manipulation in a somatosensory perceptual detection task. A** Signal detection theory model: According to SDT, valid information about stimulus probabilities changes the decision criterion c, while sensitivity Dprime should not be affected. **B** Stable environment: Participants were presented with a valid probability cue (low or high) at the beginning of each block.

Each block consisted of 12 trials, with the proportion of near-threshold and "catch" (i.e., no stimulus) trials according to the probability cue. **C** Volatile environment: Participants were presented with a probability cue (orange or blue circle) at the beginning of each trial. S signal, N noise.

## Results

### Behavioural criterion shifts in correspondence to stimulus probability

We tested the effect of stimulus probability on somatosensory near-threshold detection in two separate studies (Fig. 1A) The first study ($n = 43$) informed participants about stimulus probability at the beginning of each block consisting of 12 trials (stable environment, Fig. 1B). The stimulus onset within a trial was cued by a colour change of the fixation cross. The second study ($n = 39$) provided information about stimulus probability at the beginning of each trial (volatile environment, Fig. 1C). In both studies, signal detection theoretic (SDT) analyses verified that participants adjusted their criterion to report a stimulus based on the instructed stimulus probability, i.e., the criterion was lower (more "liberal") in the high-probability condition (mean criterion change high vs. low stimulus probability: $\Delta_{\text{crit. stable}} = -0.21$, W = 172, $p < 0.0001$, Cohen's d = 0.72, bootstrapped 95% confidence interval of median difference $[-0.258, -0.109]$; $\Delta_{\text{crit. volatile}} = -0.21$, W = 91, $p < 0.0001$, Cohens' d = 0.84, $[-0.286, -0.114]$, Fig. 2A, Biv). Participants reported that they perceived a stimulus more often in signal trials (mean hit rate (HR) change high minus low stimulus probability: $\Delta_{\text{HR stable}} = 0.07$, W = 151, $p < 0.001$, Cohen's d = 0.67,

$[0.021, 0.077]$; $\Delta_{\text{HR volatile}} = 0.08$, W = 69, $p < 0.0001$, Cohen's d = 0.98, $[0.060, 0.111]$) and noise trials (trials without a stimulus) (mean false alarm rate (FAR) change high minus low stimulus probability: $\Delta_{\text{FAR stable}} = 0.06$, W = 185, $p < 0.001$, Cohen's d = 0.62, $[0.007, 0.055]$; $\Delta_{\text{FAR. volatile}} = 0.03$, W = 169, $p = 0.002$, Cohen's d = 0.52, $[0.004, 0.038]$) under the high as compared to the low probability condition (Fig. 2Ai & ii, Bi & ii). Stimulus sensitivity (measured by the SDT parameter *Dprime*) did not significantly differ between conditions in both studies (mean *Dprime* change high minus low stimulus probability: $\Delta_{\text{Dprime stable}} = -0.06$, W = 391, $p = 0.329$, Cohen's d = 0.18, $[-0.186, -0.020]$, $\Delta_{\text{Dprime volatile}} = 0.02$, W = 361, $p = 0.686$, Cohen's d = 0.07, $[-0.160, 0.174]$, Fig. 2Aiii, Biii). Analysis of confidence responses further confirmed that participants utilised stimulus probability to inform their decision-making process. Specifically, confidence ratings in correct trials were compared based on the congruency between the probability cue and the response on each trial. We expected participants to be more confident in their yes decisions in high stimulus probability trials and no responses in low stimulus probability trials (congruent trials). Indeed, participants reported significantly higher confidence in trials that were congruent with their response compared to incongruent trials (mean confidence congruent minus incongruent correct trials: $\Delta_{\text{conf. stable}} = 0.08$,

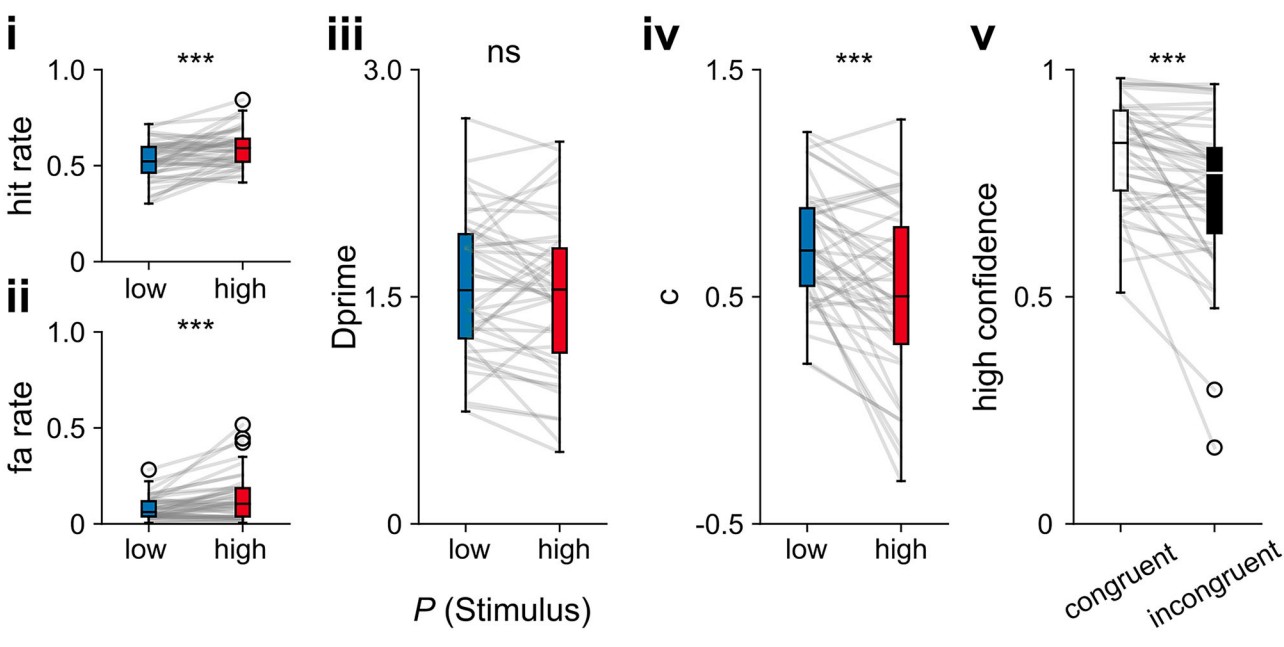

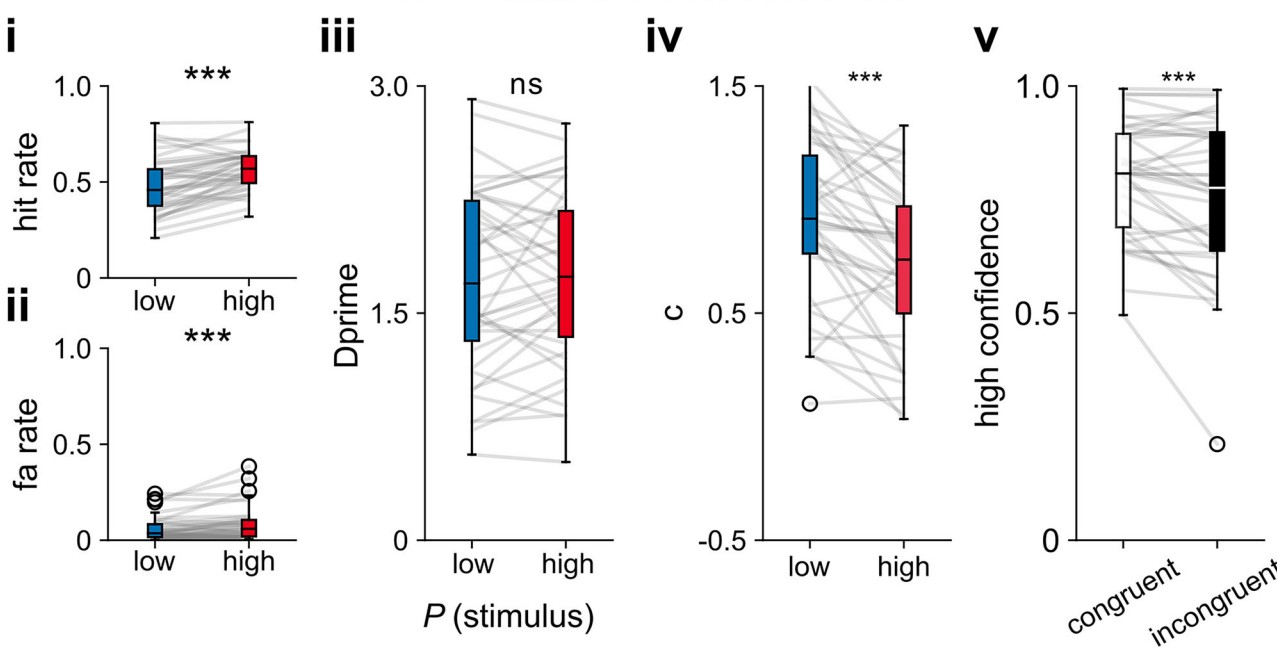

**Fig. 2 | Experimentally controlled stimulus expectations shift detection thresholds in stable and volatile probability environments. A** Signal detection theoretic analysis stable environment (**A**, n = 43 participants) and volatile environment (**B**, n = 39 participants). Participants had a higher hit rate (**i**) as well as a higher false alarm rate (**ii**) in the high expectation condition, with no significant difference in Dprime (**iii**). The criterion c was significantly more conservative in the high-probability condition (**iv**). The mean of high confidence ratings in correct trials was higher in trials in which the response matched the participants' expectations (congruent trials) (**v**). Supplementary Fig. 1.1–1.4. Box plots depict median and interquartile range, while whiskers show minimum and maximum values. Significance: ***$p < .001$, **$p < .01$, *$p < .05$. ns not significant.

W = 92, $p < 0.0001$, Cohen's d = 0.70, [0.026, 0.079]; $\Delta_{conf. volatile}$ = 0.04, W = 188, $p = 0.005$, Cohen's d = 0.51, [0.003, 044], Fig. 2Av, Bv). Notably, the excluding three participants with very high false alarm rates (>40%) in the high stimulus probability condition did not change the statistical outcomes in the stable context (Suppl. Fig. 1.1). Participants were explicitly informed about stimulus probability and trial randomisation. We hypothesised that the previous choice would be more informative in a stable environment, as participants only had to keep in mind the last few trials (12 trials) compared to the whole experimental block (120 trials) in the volatile design (Suppl. Fig. 1.2). We fitted separate signal detection theoretic generalised linear mixed models (GLMMs) with a binomial outcome variable and a probit link function for each stimulus probability environment. Each model investigated the relationship between the explicit, stimulus probability-induced bias and the implicit, choice-history bias. The basic

signal detection theoretic regression model predicts the detection response (detected or undetected) on each trial based on the stimulus (signal or noise) and the stimulus probability (high or low). The model incorporates an interaction term between stimulus and stimulus probability, representing changes in sensitivity resulting from the probability manipulation. The weight of the stimulus probability regressor represents the influence of the stimulus probability cue on the criterion c.

### Response history bias interacts with stimulus probability in a stable probability environment

To investigate how previous choices influence current choices, we included the previous detection response as a regressor in our model, which improved the model fit in both probability environments ($\Delta_{AIC\ stable} = -229$, $X2(5) = 229.83$, $p < 0.001$; $\Delta_{AIC\ volatile} = -80$, $X2(5) = 90.94$, $p < 0.001$). Overall, participants showed a tendency to repeat their previous choice ($\beta_{stable} = 0.117$, $p = 0.021$, [0.018, −0.216]; $\beta_{volatile} = 0.177$, $p < 0.001$, [0.092, 0.261]). Importantly, even after controlling for the effect of the previous choice, stimulus probability still accounted for a significant amount of variance in the detection response ($\beta_{stable} = 0.141$, $p = 0.010$, [0.034, 0.248]; $\beta_{volatile} = 0.180$, $p = 0.001$, [0.071, 0.298]). Finally, we fit a model that included the interaction between the previous choice and stimulus probability, which improved the model fit in the stable but not in the volatile context ($\Delta_{AIC\ stable} = -13$, $X2(1) = 14.91$, $p < 0.001$; $\Delta_{AIC\ stable} = +2$, $X2(1) = 0.08$, $p = 0.777$). Specifically, the model fitted with the data from the stable environment showed a significant interaction between the stimulus probability and the previous choice regressor ($\beta_{stable} = 0.163$, $p < 0.001$, [0.080, 0.245]). Post hoc tests showed a significant influence of the previous choice on the detection response both in the high and low stimulus probability conditions, but a weaker effect in the low probability condition (conditioned on low stimulus probability and previous yes response: $\beta_{stable} = 0.058$, $p = 0.021$, [0.002, 0.114]; high stimulus probability and previous yes response: $\beta_{stable} = 0.140$, $p < 0.001$, [0.092, 0.188], FDR-corrected, Suppl. Fig. 1.3 visualises the model-free interaction effect of criterion (c) and previous choice, model summaries in Suppl. Tables 1, 2).

A key feature of the stable environment is the difference in the previous response distributions between the probability blocks. While in both environments the instructed probability matched the actual stimulus probability ("valid cues"), the randomisation of probability cues within the volatile environment effectively balanced the distributions of previous "yes" and "no" responses (Suppl. Fig. 1.4).

Next, we fitted the interaction model from the stable environment separately for signal and noise trials, to investigate whether the interaction effect was driven by the unequal amount of signal and noise trials within the probability conditions. Both the 'signal trials only' model ($\beta = 0.182$, $p = 0.004$, [0.059, 0.305]), as well as the 'noise trials only' model ($\beta = 0.213$, $p = 0.027$, [0.024, 0.403]) showed a significant interaction between the choice-history induced bias and the stimulus probability-induced bias. The model fitted with trials that previously contained a signal confirmed an interaction between stimulus probability and previous response ($\beta = 0.149$, $p = 0.021$, [0.022, 0.275]). Interestingly, the interaction effect was no longer statistically significant for trials that followed noise ($\beta = 0.029$, $p = 0.742$, [−0.146, 0.205]).

### Lower pre-stimulus beta power in distinct areas associated with bias introduced by stimulus probability and previous choice

After having confirmed the behavioural effects of the explicit stimulus probability manipulation both on detection as well as confidence responses, we set out to determine the neural correlates of both explicit and implicit biases on somatosensory near-threshold detection. Our initial analysis in sensor space focused on low-frequency oscillations in the pre-stimulus window. To determine the somatosensory region of interest, we selected the EEG channel that showed the strongest early somatosensory stimulus-evoked response (i.e. here assessed by the P50 component at around 50 ms post-stimulus). In both studies, this was channel CP4, which is located over

centro-parietal areas contralateral to the somatosensory stimulation site (Suppl. Fig. 2.1). Next, we computed time-frequency representations in the pre-stimulus window in the frequency range from 3 to 35 Hz for both environments. A threshold-free cluster-based permutation test in sensor space showed no significant cluster for the contrast between high and low stimulus probability trials (Fig. 3Ai) in the stable environment (minimum cluster $p$ value: 0.087, peak $t$-value: 2.035, Cohen's d = 0.14,) but a significant cluster in the volatile environment, most likely driven by lower power in the beta range in the high stimulus probability condition (minimum cluster $p$ value: <0.001, peak $t$-value: 5.312, Cohen's d = 0.11, Fig. 3Bi and Suppl. Fig. 2.2 for a shorter pre-stimulus window). The $t$-values were most negative in the beta frequency band, suggesting that beta band modulations are forming the cluster. A cluster-based permutation test contrasting pre-stimulus power in detected (hits) and undetected signal trials (misses) showed lower alpha power immediately before detected trials in the stable environment, independent of the probability condition. Lower beta power was only clearly present in the high stimulus probability condition. In the low probability condition, beta (above 18 Hz, 100 ms before stimulation) showed no strong modulation (Suppl. Fig. 2.3Ai, ii, stable environment: high probability hit minus miss contrast: minimum cluster $p$ value < 0.001, peak $t$-value: 9.139, Cohen's d = 0.21, low probability hit minus miss contrast: minimum cluster $p$ value < 0.001, peak $t$-value: 8.286, Cohen's d = 0.18). In the volatile environment, beta power decreased before hits in the high-probability condition. There was no significant cluster in the low probability condition (Suppl. Fig. 2.3Bi, ii, volatile environment high probability hit minus miss contrast: minimum cluster $p$ value < 0.001, peak $t$-value: 3.435, Cohen's d = 0.09, low probability hit minus miss contrast: minimum cluster $p$ value = 0.063, peak $t$-value: 6.352, Cohen's d = 0.16). While these data should be interpreted cautiously, the finding of a decrease in beta power (hits versus misses), especially in the high probability trials, supports the idea that beta band power modulations are crucial for implementing explicitly induced criterion changes (Suppl. Fig. 2.3Bi, ii). Next, we reconstructed pre-stimulus beta power using a beamforming approach. After having validated our beamformer on post-stimulus data (Suppl. Fig. 2.4), we contrasted pre-stimulus beta [−700 to −100 ms] power for high minus low stimulus probability (visualisation of t-values in each vertex obtained by a permutation t-test) and located the strongest modulation in the postcentral gyrus for both environments (Fig. 3Ai, Bi). Next, we aimed to identify the effect of previous choices in the same pre-stimulus window. Threshold-free cluster-based permutation testing in the pre-stimulus window showed a significant cluster for the previous response contrast in the stable environment (minimum cluster $p$ value: 0.040, peak $t$-value: 7.766, Cohen's d = 0.17, Fig. 3Ai), but no significant cluster in the volatile environment (minimum cluster $p$ value: 0.378, peak $t$-value: 0.356, Cohen's d = 0.10, Fig. 3Bii). The strongest power modulation for the previous response contrast was in the secondary somatosensory cortex for the stable environment and in the posterior parietal cortex for the volatile environment.

### Pre-stimulus beta power predicts criterion change in both environments and interacts with the previous response in the stable environment

The analysis so far showed a modulation in the lower beta band power before stimulation onset in distinct cortical areas. Next, we investigated how the pre-stimulus beta power modulations related to the behavioural outcomes of the stimulus probability manipulation. Therefore, we averaged pre-stimulus beta power [15–25 Hz] in the pre-stimulus window (−700 to −100 ms) over voxels in source space which were among the 10% of voxels with the most negative $t$-values for the probability contrast. Note that in both environments, the strongest beta modulation was in the postcentral gyrus. Behavioural modelling showed that the probability manipulation led to a change in participants' decision criteria in both studies, with an interaction between the previous response and stimulus probability only in the stable environment. If pre-stimulus beta power is a neural correlate of the experimental manipulation of stimulus probabilities, i.e., reflecting the change in criterion, it should also predict the change in detection rates (for

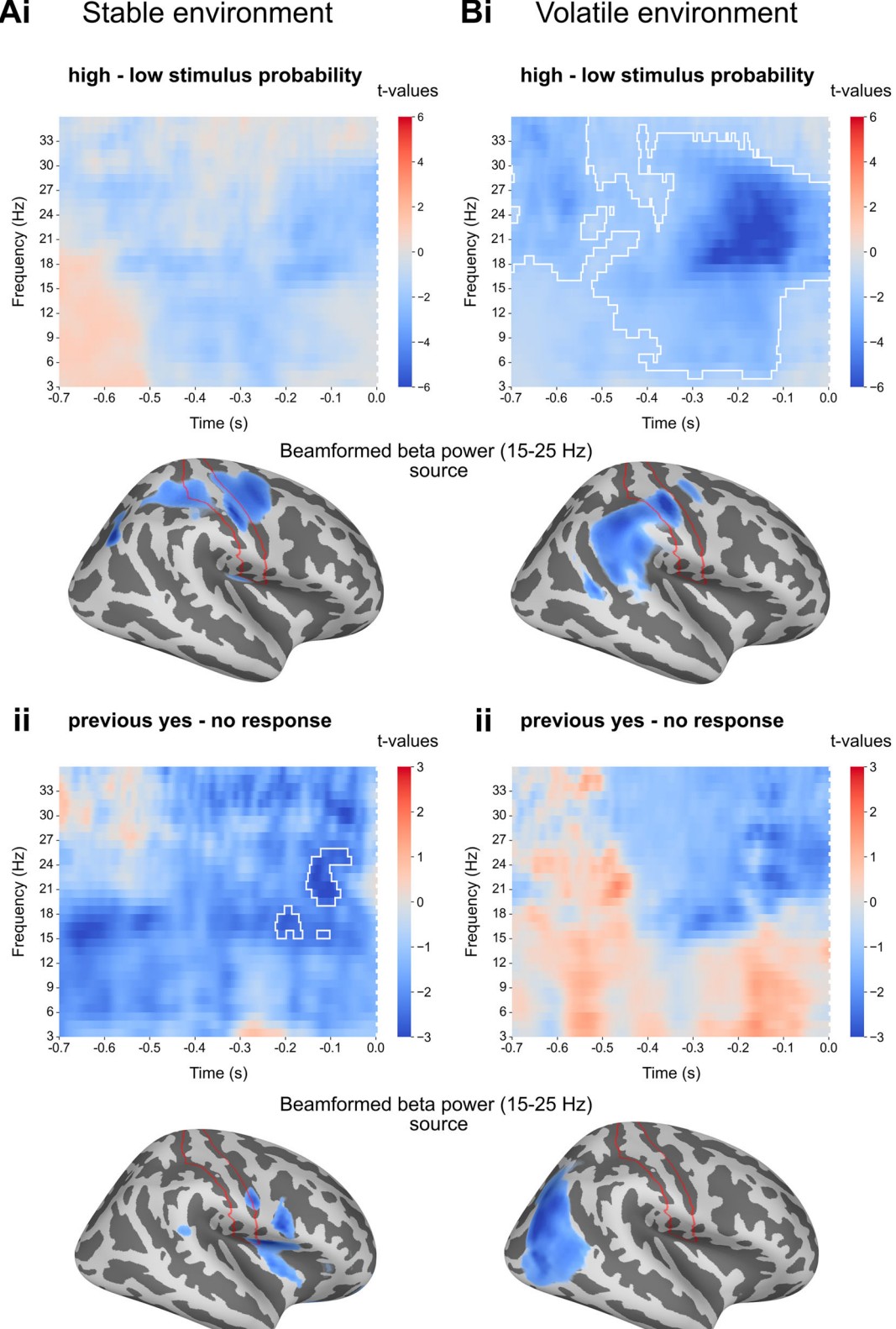

signal and noise trials) in both studies and mimic the interaction with the previous choice in the stable environment. For the single-trial analysis, we calculated pre-stimulus beta power averaged over the most discriminative voxels for the stimulus probability contrast. Brain-behaviour modelling confirmed that stimulus probability could be decoded from pre-stimulus beta power in somatosensory areas ($\beta_{stable} = -0.030$, $p = 0.009$, [$-0.052$,

$-0.008$]; $\beta_{volatile} = -0.0480$, $p < 0.001$, [$-0.070$, $-0.026$]), with lower power predicting higher stimulus probability. Participants responded more often that they detected a stimulus both in signal and noise trials after lower pre-stimulus beta power, which suggests that pre-stimulus beta power is a neural correlate of the criterion change ($\beta_{stable} = -0.099$, $p = 0.001$, [$-0.159$, $-0.039$]; $\beta_{volatile} = -0.175$, $p < 0.001$, [$-0.249$, $-0.101$]; Fig. 4Ai). In the

**Fig. 3 | Lower pre-stimulus beta power in high probability trials and after a previous yes response in distinct cortical areas. Ai** In the stable environment (*n* = 43 participants), a threshold-free cluster permutation test showed no significant cluster for the difference between high and low stimulus probability in the pre-stimulus window (minimum *p*-value: .087). A *t*-test on the source reconstructed pre-stimulus beta power contrast locates the strongest modulation (most negative *t*-values) in the postcentral gyrus. **Bi** In the volatile environment (*n* = 39 participants), a threshold-free cluster permutation test detected a significant cluster. The effect was most pronounced around stimulus onset in the beta band, with lower power in the high stimulus probability condition (minimum *p* value = 0.001). The beta power

source was localised in the postcentral gyrus. **Aii** A significant cluster was found in the pre-stimulus window for the contrast between previous yes - previous no responses, which was most likely driven by lower beta power after a previous yes response (minimum *p* value: 0.040) and for which the strongest beta modulation originated from the secondary somatosensory cortex (SII). **Bii** No significant cluster for the contrast of previous choices was found in the volatile environment (minimum *p* value: 0.378). The strongest beta modulation originated from the posterior-parietal cortex. The area marked in red highlights the postcentral gyrus. The 10% most negative *t*-values are highlighted in source space, with darker colours representing more negative values. Suppl. Fig. 2.1–2.4.

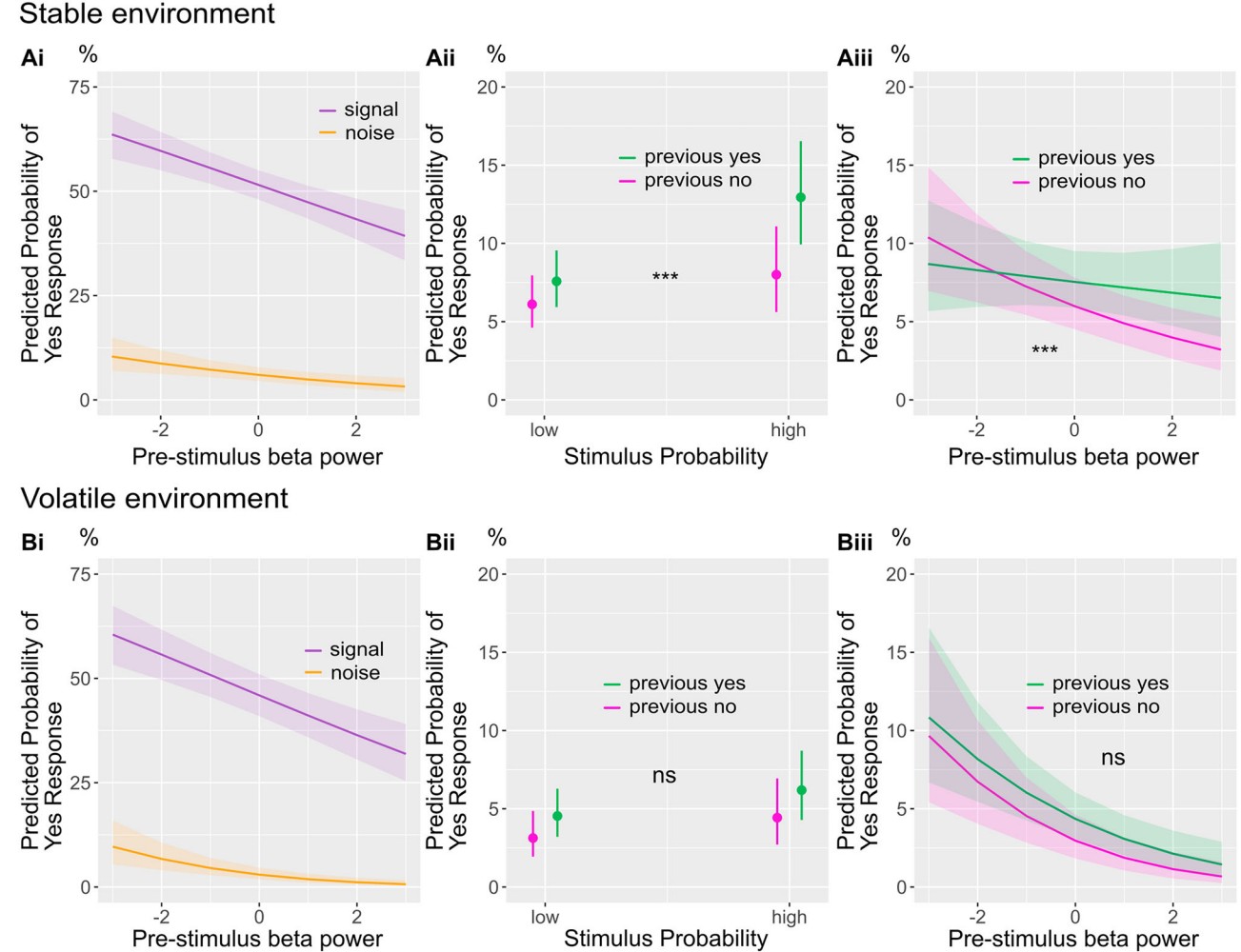

**Fig. 4 | Pre-stimulus beta power predicts criterion change and interacts with stimulus probability in the stable environment. Ai** The probability for a detection response decreases with higher pre-stimulus beta power for both signal and noise trials in the stable environment (*n* = 43 participants). **Bi** The probability for a stimulus report decreases with higher pre-stimulus beta power for both signal and noise trials in the volatile environment (*n* = 39 participants). **Aii** The effect of the previous response depends on the stimulus probability condition in the stable environment (significant interaction). **Bii** Participants respond more often that they detect a stimulus after previous yes responses in both probability conditions (no

significant interaction) in the volatile environment. **Aiii** In the stable environment, the previous response interacts with pre-stimulus beta power regarding its relationship with the probability of a yes response such that with a previous "no response" the probability of a "yes response" decreases while there is no effect after a previous yes response. **Biii** The probability for a yes response decreases with increasing beta power, independent of the previous response. Significance levels: ***p* < 0.001, ***p* < 0.01, **p* < 0.05, shaded areas and error bars show the 95% confidence interval.

stable environment, the best-fitting model included an interaction between the previous response and beta power ($\Delta_{AIC} = -17.58$, X2(2) = 21.578, *p* < 0.001). The interaction between beta power and previous choices was such that for low beta power, the effect of the previous choice on the detection response was reversed in comparison to the relationship for high beta power (Fig. 4Aii). The best-fitting model in the volatile environment did not include an interaction between previous choices and stimulus probability ($\Delta_{AIC} = 2.65$, X2(2) = 1.35, *p* = 0.508), participants responded

more often that they detected a stimulus after low beta power and after previous yes responses (Fig. 4 Bii, model summaries in Suppl. Tables 3, 4).

**Pre-stimulus beta power mirrors the congruency effect on confidence ratings**

To reinforce the significance of pre-stimulus beta power as a neural correlate of stimulus expectations, we aimed to validate its ability to account for the congruency effect on confidence ratings observed in the behavioural model.

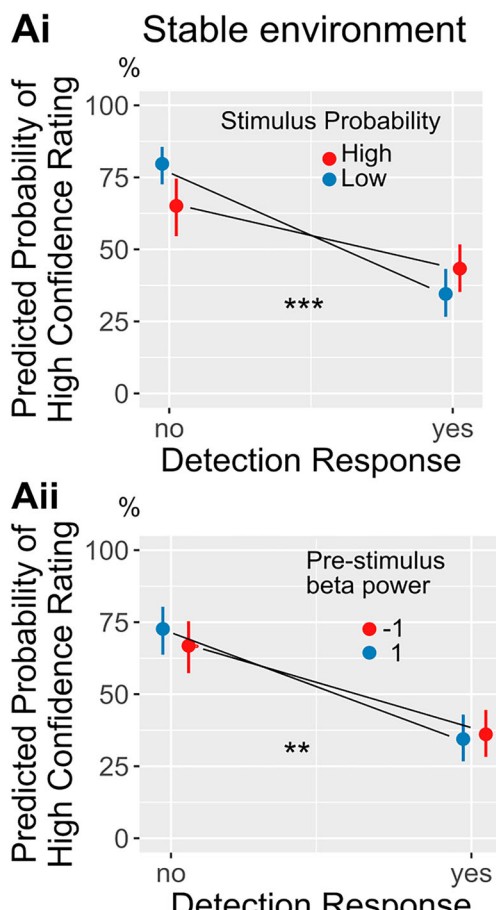

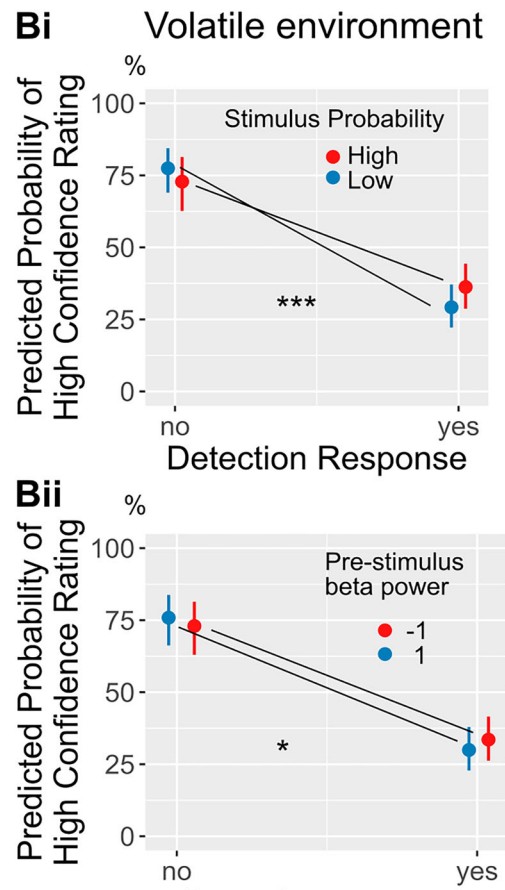

**Fig. 5 | Pre-stimulus beta power predicts the congruency effect on confidence ratings. Ai** The congruency effect on confidence between response and stimulus probability in the stable environment ($n = 43$ participants): participants are more confident in no responses in the low probability condition and more confident in yes responses in the high probability condition. **Bi** The congruency effect on confidence between response and stimulus probability in the volatile environment ($n = 39$ participants): participants are more confident in no responses in the low probability condition and more confident in yes responses in the high probability condition. **Aii**

Significant congruency effect on confidence between response and beta power in the stable environment: beta power does predict the confidence for both yes and no responses, conditional on the probability condition. **Bii** Significant congruency effect on confidence between response and beta power in the volatile environment for yes responses: lower beta power increases confidence in yes responses. The error bars show the 95% confidence interval. Significance levels $***p < 0.001$, $**p < 0.01$, $*p < 0.05$.

Congruency was defined as the alignment between stimulus probability and the detection response. For example, a "yes" response in the high probability condition would be considered a probability-congruent response. Participants were more confident in correct trials after a no response and following a previous high confidence rating. Crucially, pre-stimulus beta power mimicked the response congruency effect on confidence: Participants were more confident in yes responses in the high probability condition compared to the low probability condition ($\beta_{stable}$ probability*response = 0.673, $p < 0.001$, [0.456, 0.899]; $\beta_{volatile}$ probability*response = 0.342, $p < 0.001$, [0.177, 0.506]; Fig. 5Ai, Bi) and analogously in trials with low pre-stimulus beta power (a feature of the high probability condition) compared to trials with high pre-stimulus beta power ($\beta_{stable}$ beta*response = −0.106, $p = 0.002$, [0.174, 0.038]; $\beta_{volatile}$ beta*response = −0.095, $p = 0.019$, [0.175, 0.016]; Fig. 5Aii, Bii). The inverse behaviour was seen for No-responses (Suppl. Tables 5, 6).

### Distinct pre-stimulus beta power sources partially mediate the effect of stimulus probability and the previous response on somatosensory near-threshold perception

Finally, a neural correlate of explicit and implicit biases should partially mediate the effect of stimulus probability and choice-history biases on the detection response. We used causal inference methods[51,52] to determine the indirect effect of either stimulus probability or previous choice on detection

mediated by pre-stimulus beta power (Fig. 6A, B; c' path). We used linear mixed-effects models to estimate the "a path", which represents the direct effect of either stimulus probability or previous choice on pre-stimulus power. The "b path" is represented via the effect of pre-stimulus beta power on the detection response. Pre-stimulus beta power in probability discriminative areas partially mediated the effect of stimulus probability on detection, with a less pronounced effect in the stable environment ($\beta_{stable}$ prop. mediated = 0.009, $p = 0.010$, [0.002, 0.020]; $\beta_{volatile}$ prop. mediated = 0.032, $p < 0.001$, [0.008, 0.040]; Fig. 6Ai, Bi). The proportion mediated for the previous choice via beta power was similar in both environments ($\beta_{stable}$ prop. mediated = 0.012, $p < 0.001$, [0.005, 0.02]; $\beta_{volatile}$ prop. mediated = 0.015, $p = 0.004$, [0.001, 0.040]; Fig. 6Aii, Bii, model summaries in Suppl. Tables 7, 8). In summary, the mediation analysis supports the role of pre-stimulus beta power in implementing explicit and implicit biases in distinct cortical areas.

### Discussion

Here, we investigated the neural mechanisms underlying biases in somatosensory perception in a stable and volatile stimulus probability environment. By manipulating the expectation of stimulus occurrence via visual cues in blocks or on a single-trial level in two separate studies, we show that participants adjust their perceptual decision criterion and confidence ratings based on the cued stimulus probability. Next to the

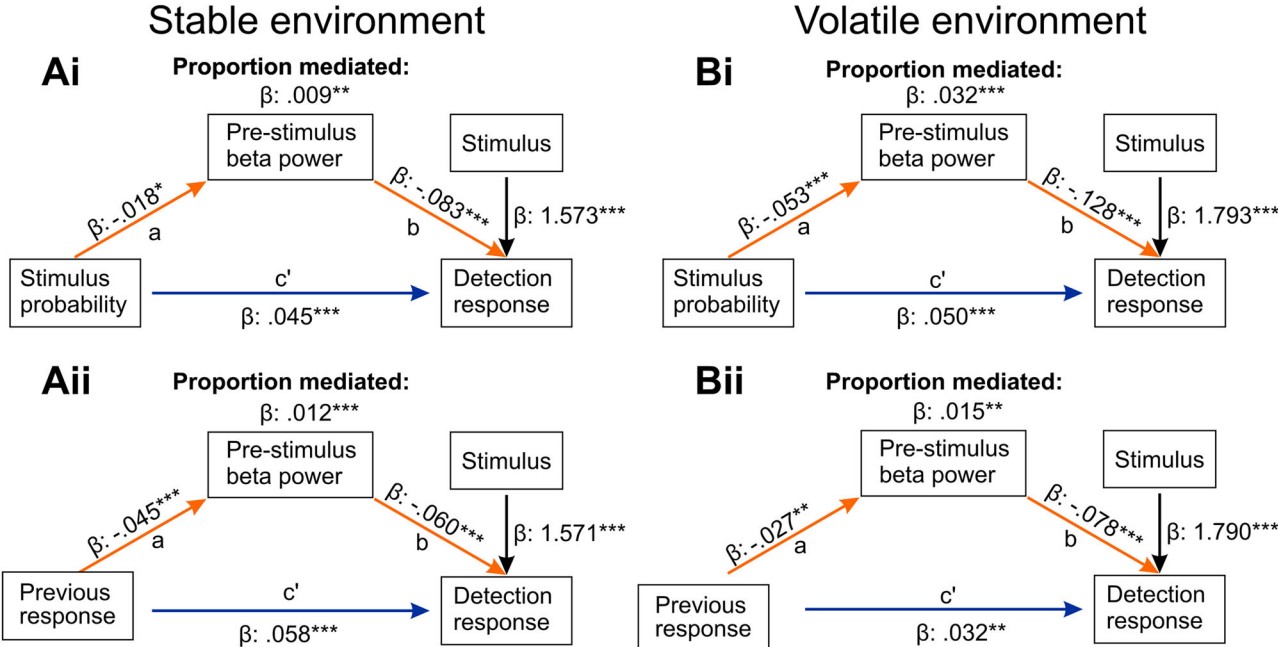

**Fig. 6 | Distinct pre-stimulus beta power sources mediate the effect of stimulus probability and previous choice on the detection response. A** Partial mediation of probability (**i**) and previous response (**ii**) by distinct beta power sources on detection in the stable environment ($n$ = 43 participants). **B** Partial mediation of probability (**i**) and previous response (**ii**) by distinct beta power sources on detection in the volatile environment ($n$ = 39 participants). Significance levels: ***$p < 0.001$, **$p < 0.01$, *$p < 0.05$.

explicit biases, participants also exhibited strong previous choice biases. In EEG recordings, we identified pre-stimulus beta power in distinct brain areas as neural correlates of explicit and implicit biases: Pre-stimulus beta power in the postcentral gyrus predicted single-subject criterion changes and partially mediated the effect of stimulus probability on the detection response. Pre-stimulus beta power also reflected a key feature of the stable probability environment: an interaction with the previous choice. Single-trial beta power modulations also mimicked the impact of the explicit bias on confidence ratings in both environments. In the volatile probability environment, the previous choice bias was reflected in pre-stimulus beta power in posterior parietal areas. In the stable probability environment, the strongest previous choice beta power modulation was localised in the secondary somatosensory cortex. Both sources partially mediated the effect of the previous response on the current response. In summary, we suggest that pre-stimulus beta power in distinct cortical areas implements explicit and implicit biases.

### Implicit and explicit priors
In the first experiment, participants were informed about stimulus probability in a block-wise manner, which arguably is a more ecologically valid design. However, since the probability cue was valid, the probability blocks contained an unequal amount of signal and noise trials and thus the response distributions (and hence previous choice frequencies) between the low and high probability conditions were unbalanced; for this reason, in the second experiment, the probability cue was provided before each trial resulting in a balanced previous choice distribution. In both environments, the probability cues led to changes in criterion, as previously shown for the visual domain[4,24,25], and the previous response significantly predicted the detection response in the current trial. Those results support previous findings on response history biases even in designs where the stimulus order is randomised[53,54]. Furthermore, in both environments, participants gave higher confidence ratings in trials where their expectation matched their response[55]. A difference between the two study designs is that in the stable environment, the last trial is more informative for the current subjective belief about stimulus probability within the same block (Suppl. Fig. 1.2). This difference was also reflected

in our model-based analysis of the participants' behaviour: The best model for the stable environment included an interaction term between the previous response and stimulus probability. Participants only relied on their previous responses in the high probability blocks, where they encountered mainly near-threshold signals with high uncertainty. In summary, behavioural modelling confirmed that explicit and implicit biases shape somatosensory near-threshold detection and interact in stable probability environments.

### Somatosensory pre-stimulus beta oscillations as a neural correlate of explicit biases
After confirming the behavioural relevance of stimulus probability on somatosensory detection and confidence in both environments, we examined how stimulus probability and previous choice impacted neural activity before stimulation onset. Time-frequency representations for the contrast of high minus low stimulus probability in the pre-stimulus window showed a significant negative cluster, with the most profound effect in the beta band for the volatile environment. In the stable environment, participants had to remember the probability cue throughout 12 trials, potentially leading to weaker effects of the cue in the pre-stimulus period. Conversely, in the volatile environment, the probability cue before each trial appears to "dominate" the pre-stimulus characteristics over the effect of the previous trial. Contrasting hits and misses in the pre-stimulus window supported this idea. In the stable and volatile environment, the contrast showed significantly lower beta power immediately before stimulation onset when the expectation for stimulus occurrence was high. There was no significant cluster for hits minus misses when the expectation for stimulus occurrence was low. Importantly, the localised sources of the strongest beta power modulation for the probability manipulation were in similar locations within the postcentral gyrus.

Our finding of beta modulation reflecting explicit probability bias is consistent with the assumed role of beta for top-down modulation of perceptual decisions and more generally as a neural correlate of top-down expectations[56,57]: For the somatosensory domain, the study by van Ede et al. (2010) suggested pre-stimulus beta power as a potential neural correlate of tactile expectations[58] with lower power before an expected somatosensory

stimulus. Ede and colleagues controlled for trial history and showed that pre-stimulus beta power implements top-down expectations. This finding is consistent with a study investigating painful stimuli by ref. 59. The authors show decreases in alpha and beta power in the pre-stimulus window in trials with a high expectation for a painful stimulus in somatosensory channels. Weisz et al.[60] identified a pre-stimulus alpha/beta modulation in the contralateral secondary somatosensory cortex during a near-threshold somatosensory detection task. They interpreted these modulations with a change in the starting point for perceptual decisions. Importantly, this does not speak against the proposed mechanism for criterion changes via shifts in baseline sensory excitability. Instead, we suggest that pre-stimulus beta oscillations mediate stimulus probability on tactile detection, potentially via changing baseline activity in sensory areas before stimulus onset. The crucial difference to previous studies is the explicit manipulation of the criterion via expectations in contrast to spontaneous modulations of the baseline, which were mainly related to alpha power[21]. Notably, lower beta power in somatosensory regions has also been associated with increased detection rates[18]. We propose that pre-stimulus beta power modulations are observed in tasks where top-down influences dominate spontaneous baseline shifts[29,30]. Supporting this idea, beta power modulations related to top-down processing have been observed in regions typically dominated by alpha activity[61].

One possible explanation for the varying contributions of alpha and beta power in somatosensory perception could be the dynamic reconfiguration of brain networks. Recent work by Sharma and colleagues[62] suggests a gradual shift from parietal to frontal networks in the pre-stimulus window of a somatosensory detection task. Based on those findings, we suggest that studies focused on localised brain dynamics may only capture transient states within a broader, interconnected network. Lower pre-stimulus alpha and beta power in sensory areas during trials with a reduced perceptual threshold may reflect efficient information routing within different states of the same network.

We observed lower pre-stimulus beta power in trials with high stimulus probability, which were behaviourally linked to a lower detection threshold. While previous studies associate higher pre-stimulus beta power with predictable stimulus onsets[63], our findings cannot be explained by temporal prediction, as stimulus timing was equally predictable across conditions. An alternative interpretation relates to the role of beta oscillations in maintaining cognitive states. Though not designed to investigate working memory, our results align with the idea that higher beta power reflects a resistance to change[29], as trials with higher pre-stimulus beta were associated with a more conservative detection threshold. Another perspective comes from research on statistical learning. Bogarts et al. observed lower pre-stimulus beta power within learned sequences compared to sequence transitions[64]. Our paradigm prevented learned predictions through pseudo-randomisation of stimuli and probability cues. The proposed role of beta power in encoding uncertainty is intriguing; however, the current design of this study does not allow for strong conclusions on their relationship. We conducted two separate studies with two different samples of participants, which allowed us to confirm the behavioural results in an independent sample. A follow-up study could leverage a longitudinal design to explore the temporal stability of both externally induced biases and implicit previous-choice biases. This would address whether explicit and implicit biases are better characterised as traits[65] or states. We localised the pre-stimulus beta power modulation in somatosensory areas in both studies, but we refrain from concluding that this is evidence for a change in subjective experience. It has been shown that a reproduction task is necessary to distinguish between changes in subjective experience and decision-related processes[66].

Our results show that beta power in the primary somatosensory cortex plays a key role in implementing explicit biases; this leaves open how the previous choice bias is implemented.

### Pre-stimulus posterior parietal and SII beta oscillations as a neural correlate of implicit biases

Time-frequency representations for the previous response contrast revealed a significant cluster in the beta band for the stable environment with lower power after previous yes responses. Source reconstruction of pre-stimulus beta power for the previous response contrast highlighted the posterior parietal cortex as the source of the strongest difference between previous responses in the volatile environment. Neural correlates of previous choice biases in nonhuman primates have been located in the frontal and posterior parietal cortex (PPC)[67,68], while a recent study in humans suggested pre-stimulus gamma power in the parietal cortex as a neural marker of previous choice biases[54]. Consistently, our findings emphasise the role of posterior parietal brain areas mediating the effect of the previous choice on the current choice via pre-stimulus beta power.

Interestingly, in the stable environment, the beta power source for the previous response contrast highlighted another area, i.e., the secondary somatosensory cortex (SII). As outlined earlier, the subjective relevance of the previous response and thus the "cognitive process" underlying the previous choice is expected to be different in the stable environment. A study by ref. 69 in nonhuman primates revealed that neurons in SII encoded both past and present sensory information during a frequency discrimination task. Primates had to remember the previous stimulus to compare it with the current one. In our task, participants were instructed to base their responses solely on the current stimulus while remembering the stimulus probability. However, in a stable probability environment, human observers are likely to keep track of recently encountered stimuli and base their decisions not only on the current stimulus but also on past stimuli. We suggest that SII integrates information from past stimuli not only in discrimination tasks but also in detection tasks with known stimulus probabilities. Future studies should directly test this idea by comparing neural modulations in discrimination intervals with those observed in inter-trial intervals. This idea is supported by findings from a rodent tactile working memory task[70], which demonstrated that SII enables task information to persist across different behavioural states. Given the evidence from previous studies and our findings on the roles of PPC and SII, we tentatively conclude that both areas are involved in the neural implementation of previous choice biases, with each region's role being more strongly emphasised in stable and volatile probability environments.

### Pre-stimulus beta power predicts explicit and implicit biases, mimics the congruency effect on confidence and mediates both stimulus probability and previous response bias in distinct brain areas

To investigate the relationship between pre-stimulus power and behavioural outcomes, we added pre-stimulus source power averaged over the most discriminative voxel for either probability or previous responses in our perceptual models. The results confirmed the role of pre-stimulus beta power in predicting responses for signal and noise trials; pre-stimulus beta power also mimicked the interaction of the previous response with the stimulus probability in the stable environment. Finally, we showed that somatosensory beta power partially mediated the effect of stimulus probability and previous choice on the detection response. While the proportion mediated was small, the effect sizes are similar to previous studies investigating brain-behaviour mediation models in perceptual decision making and oscillations[23,54]. A recent review indicates that beta oscillations, particularly beta bursts, occur across various cortical regions[28]. The authors suggest that beta oscillations act as spatiotemporal filters, thereby controlling information flow throughout the brain via brief periods of functional inhibition. We show distinct beta sources for implicit and explicit biases in the pre-stimulus window that shape somatosensory perception.

The observed effect of higher confidence in response-congruent trials could be replicated by a model that included pre-stimulus beta power as a neural correlate of stimulus probability. Those results illustrate the intricate interplay between pre-stimulus power and confidence, as it is not the absolute level of beta power that determines confidence. Both high and low beta power before a stimulus can lead to a high confidence perceptual decision—it depends on the response and whether it matches the expected stimulus. Earlier studies have emphasised a negative association between pre-stimulus alpha power and confidence[71,72]. Extending this perspective,

ref. 15 demonstrated in a somatosensory discrimination task that the correlation between pre-stimulus alpha power and confidence varies depending on accuracy. The results of our study suggest that the relationship between pre-stimulus beta power and confidence depends on the congruency between the response and the expectation, further supporting the idea that beta oscillations are crucial for implementing top-down biases in sensory areas.

## Limitations

The computational models used assume a linear relationship between pre-stimulus power and the detection response[73]. Consequently, nonlinear effects between power and behavioural outcomes cannot be detected by the models used in this study. Finally, the electrical stimulation used in our study, although widely used in research, is not a naturalistic stimulus.

## Data availability

The raw behavioural and electrophysiological data are publicly available on EDMOND.

## Code availability

The experimental (Matlab) and analysis code (Python and R) are publicly available on the first author's Github account (CarinaFo), and a snapshot of the code is stored on Zenodo.

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

## Acknowledgements

The authors thank Sylvia Stasch and Ramona Menger for support with behavioural and EEG data collection, and Jule Bohlen and Anastassia Loukianov for help with the preparation of the volatile environment and data collection. We further thank Anahit Babayan for the help with the preparation of the ethics proposal. C.F. wants to thank the MNE core developers for their ongoing support with EEG analysis and Python coding, especially Dan McCloy and Eric Larson. We further thank Phillip Bokiniec for his help with data visualisation. C.F. was supported by a stipend from the Einstein Centre for Neurosciences, Charité Universitätsmedizin, Berlin, while implementing the stable environment. The funders had no role in study design, data collection and analysis, decision to publish or preparation of the manuscript.

## Author contributions

C.F. and A.V. conceived the study. C.F., A.V., E.A., and M.G. designed the first study. C.F., A.V., T.S., E.P., and V.N. designed the second study. C.F., M.G., and E.P. prepared the experimental setup. C.F. analysed behavioural and neural data for both studies with help from T.S. and V.N. C.F. wrote the first draft of the manuscript with input from A.V., S.H., and E.P. reviewed the analysis code. All authors reviewed, edited and approved the final manuscript.

## Funding

## Competing interests

The authors declare no competing interests.
