## [Transparent Peer Review file · Communications Psychology]

Pre-stimulus beta power mediates explicit and implicit perceptual biases in distinct cortical areas

Corresponding Author: Ms Carina Forster

Version 0:

Decision Letter:

Dear Ms Forster,

Thank you for your patience during the peer-review process. Your manuscript titled "Pre-stimulus beta power mediates explicit and implicit perceptual biases in distinct cortical areas" has now been seen by 2 reviewers, and I include their comments at the end of this message. They find your work of interest but raised some important points. We are interested in the possibility of publishing your study in Communications Psychology, but would like to consider your responses to these concerns and assess a revised manuscript before we make a final decision on publication.

We therefore invite you to revise and resubmit your manuscript, along with a point-by-point response to the reviewers. Please highlight all changes in the manuscript text file.

Editorially, we consider it important that the direction of the beta activity modulations, as well as the prominence of beta oscillations in somatosensory decision, should be made clearer in the revised manuscript. Please also clarify the methodological choices such as sensor and time window selections per Reviewer #1's suggestion. Finally, please clarify what difference in AIC scores you considered a meaningful benchmark for better model fit and provide supporting arguments for this decision.

I am attaching an Editorial Requests Table that details critical reporting requirements for the revised manuscript. Please attend to each item and ensure your manuscript is fully compliant. If your revised manuscript is not aligned with these requests on major issues, such as those concerning statistics, it may be returned to you for further revisions without re-review.

Please submit the following items:

- Revised manuscript
- Point-by-point response to the referees' comments
- Cover letter (as a separate document)
- [Nature Research Reporting Summary](https://www.nature.com/documents/nr-reporting-summary.zip)
- [Editorial Policy Checklist](https://www.nature.com/documents/nr-editorial-policy-checklist.pdf)
- Completed Editorial Request Table (attached).

via this link: Link Redacted .

Additional guidance is available in our style and formatting guide Communications Psychology formatting guide.

Best regards,

Troby Lui

Troby Lui, PhD
Associate Editor
Communications Psychology

REVIEWER EXPERTISE:

Reviewer #1: neural oscillations, bias/previous history

Reviewer #2: neural oscillations, bias/previous history

REVIEWER REPORTS:

Reviewer #1 (Remarks to the Author):

Overall, my evaluation of the paper is positive. The results of the two experiments are convincing and timely. The suggestions for further improvements I have are focused on the theoretical framing of the paper and on clarifying some of the methodological decisions.

#1 Labels Explicit and Implicit Bias: I find the labels explicit and implicit bias rather misleading, as in cognitive science at large these typically refer to the level of awareness of the participant rather than being defined as what is or is not explicitly controlled by the experimenter.

#2 Direction of Observed Beta power modulation and integration with existing literature: The first paragraph of the discussion could be more clear about the direction of the observed effect. It is suggested that "pre-stimulus beta power in distinct cortical areas implements both explicit and implicit biases", but how should this be interpreted given that it is decreased beta-power that is found to be a neural correlate of stimulus expectations? I thought that the direction of this finding also deserved some extra attention in terms of linking to the existing literature. In contrast to the reported findings locally increased beta activity before expected sensory events has been suggested to reflect "the mobilization of neuronal populations under predictive signals" (Arnal & Giraud, 2012, TiCS; Bernasconi et al., 2011, Int. J of Psychophysiology). A recent paper which found, in line with the reported findings, decreased beta power prior to expected visual events (based on prior learning of stimulus sequences) is the one by Bogaerts, Richter, Landau & Frost (2020, J Neuroscience), which might also be interesting to discuss.

#3 Several methodological choices would benefit from additional justification:

- Sample Sizes: While the sample sizes appear reasonably large, but it is not clear how they were determined.
- Response History Bias Definition: The response history bias (also labeled implicit bias) is defined based on the previous choice (n-1). Why not include a longer choice history (e.g., n-2 or n-3)?
- Sensor Selection for Pre-Stimulus Analysis: CP4, which reveals the larger P50, was selected as the sensor of interest for the pre-stimulus time-frequency analysis. Could the authors elaborate on this choice? Wouldn't it also be reasonable to expect divergent spatial localization of pre-stimulus versus evoked brain signatures? More generally, the rationale for

selecting a single sensor yet doing source reconstruction was not entirely clear to me.

- Pre-Stimulus Beta Power Window: On page 14, it is mentioned that beta power across the entire pre-stimulus window was averaged. However, Figure 3 shows that the effect is primarily present in the final 300 ms leading up to stimulus onset (-0.3 to 0 s).

More minor Issues:

- In Figures 3 and 4, the axes are not consistent across different panels. Aligning these axes would facilitate accurate interpretation of the figures.
- It is unclear what the "beta estimate above Pre-stimulus beta power" reflects in Figure 6. Providing additional detail would help readers understand this aspect.

Reviewer #2 (Remarks to the Author):

The study explores how expectations and prior actions influence perception, focusing on pre-stimulus brain oscillations as potential markers of these biases. Two EEG experiments examined implicit biases (previous choices) and explicit biases (informed stimulus probabilities) in somatosensory detection tasks under volatile and stable contexts. Behavioral results showed decision criteria and confidence ratings were shaped by stimulus probability cues and choice history. Spectral EEG analysis revealed that pre-stimulus beta power from sensory and higher-order cortical areas predicted explicit and implicit biases, respectively, and mediated the effects of prior choices and probabilities on detection responses. This suggests pre-stimulus beta oscillations reflect neural mechanisms underlying these biases.

The results are interesting and the paper presents a thorough investigation of which factors determine oscillatory brain dynamics in relation to different types of bias. I have two points for a small revision:

- The two experiments are discussed in two different figures, and the similarities and differences between the studies are not so clear. If it would be possible to combine design and results of both studies in one figure that would be helpful to grasp the overall experimental structure.
- The current study focuses on beta oscillations, whereas many previous studies find a clear link between alpha power and decision bias. It would be interesting to read more about why beta occurs more prominently in somatosensory decisions. Relatedly, both alpha and beta is suppressed during more liberal decision making. What does it mean when an oscillation is reduced - i.e. how can it be a meaningful signal? Ole Jensen et al. have tried to answer this question in several papers and reviews, but it still remains an open question. A short discussion would be helpful for interpreting beta in this paper.

Communications Psychology is committed to improving transparency in authorship. As part of our efforts in this direction, we are now requesting that all authors identified as 'corresponding author' create and link their Open Researcher and Contributor Identifier (ORCID) with their account on the Manuscript Tracking System prior to acceptance. ORCID helps the scientific community achieve unambiguous attribution of all scholarly contributions. You can create and link your ORCID from the home page of the Manuscript Tracking System by clicking on 'Modify my Springer Nature account' and following the instructions in the link below. Please also inform all co-authors that they can add their ORCID to their accounts and that they must do so prior to acceptance.

Version 1:

Decision Letter:

Dear Ms Forster,

Your manuscript titled "Pre-stimulus beta power mediates explicit and implicit perceptual biases in distinct cortical areas" has now been seen by our reviewers, whose comments appear below. In light of their advice I am delighted to say that we are happy, in principle, to publish a suitably revised version in Communications Psychology.

We therefore invite you to revise your paper one last time to address the remaining concerns of our reviewers and a list of editorial requests. At the same time we ask that you edit your manuscript to comply with our format requirements and to maximise the accessibility and therefore the impact of your work.

EDITORIAL REQUESTS:

SUBMISSION INFORMATION:

OPEN ACCESS:

*** TRANSPARENT PEER REVIEW:** Communications Psychology uses a transparent peer review system. On author request, confidential information and data can be removed from the published reviewer reports and rebuttal letters prior to publication. If you are concerned about the release of confidential data, please let us know specifically what information you would like to have removed. Please note that we cannot incorporate redactions for any other reasons.

*** CODE AVAILABILITY:** All Communications Psychology manuscripts must include a section titled "Code Availability" at the end of the methods section. We require that the custom analysis code supporting your conclusions is made available in a publicly accessible repository at this stage; please choose a repository that generates a digital object identifier (DOI) for the code; the link to the repository and the DOI must be included in the Code Availability statement. Publication as Supplementary Information will not suffice.

*** DATA AVAILABILITY:**

Link Redacted

Best regards,

Troy Lui

Troy Lui, PhD
Associate Editor
Communications Psychology

REVIEWERS' COMMENTS:

Reviewer #1 (Remarks to the Author):

The authors responded thoroughly to all of the concerns raised in my previous review. I can recommend publication.

Reviewer #2 (Remarks to the Author):

The authors did a great job addressing my comments and I don't have any further recommendations

Reviewer #1: neural oscillations, bias/previous history

Reviewer #2: neural oscillations, bias/previous history

Editor comments not addressed within the reviewer's comments:

Finally, please clarify what difference in AIC scores you considered a meaningful benchmark for better model fit and provide supporting arguments for this decision.

We thank the editor for pointing this out. We considered a difference in AIC of greater 10 as meaningful based on the textbook by Burnham & Anderson (2004). We additionally used the `anova()` function from the stats package (3.6.2) in R to run likelihood ratio tests on the nested models.

We added the following sentences to the methods part:

p. 35, l. 810ff

We compared Akaike Information Criterion (AIC) measures of each model and considered a difference greater than 10 as a meaningful difference in model fit (Burnham&Anderson, 2004). As an independent source of model fit, we compared models using the `anova` function from the stats package (3.6.2) in R which compares nested models based on likelihood ratio tests.

REVIEWER REPORTS:

Reviewer #1 (Remarks to the Author):

Overall, my evaluation of the paper is positive. The results of the two experiments are convincing and timely. The suggestions for further improvements I have are focused on the theoretical framing of the paper and on clarifying some of the methodological decisions.

We thank the reviewer for the positive evaluation of the manuscript and the time taken to review this paper.

#1 Labels Explicit and Implicit Bias: I find the labels explicit and implicit bias rather misleading, as in cognitive science at large these typically refer to the level of awareness of the participant rather than being defined as what is or is not explicitly controlled by the experimenter.

We appreciate the reviewer's feedback regarding explicit and implicit bias. While writing the manuscript, we discussed and considered alternatives and agreed that implicit and explicit are the "least" misleading terms. An alternative would have been to refer to explicit bias as a high-level prior and choice history bias as a low-level prior (Eckert et al., 2023). However, we thought those labels implied separate processing levels, which we do not support with our results. We agree that the definition of explicit and implicit biases has to be addressed, thus we added the following paragraph to the introduction:

p. 3 l. 61ff

In the following, we define the bias induced by choice history as implicit bias, as it arises from internal processes that the experimenter does not explicitly control. It is important to note that the terms implicit and explicit, as used throughout this manuscript, do not pertain to participants' level of awareness. Rather, we label the choice history bias as implicit because it emerges without explicit information provided to the observer.

#2 Direction of Observed Beta power modulation and integration with existing literature: The first paragraph of the discussion could be more clear about the direction of the observed effect. It is suggested that “pre-stimulus beta power in distinct cortical areas implements both explicit and implicit biases”, but how should this be interpreted given that it is decreased beta-power that is found to be a neural correlate of stimulus expectations? I thought that the direction of this finding also deserved some extra attention in terms of linking to the existing literature. In contrast to the reported findings locally increased beta activity before expected sensory events has been suggested to reflect “the mobilization of neuronal populations under predictive signals” (Arnal & Giraud, 2012, TiCS; Bernasconi et al., 2011, Int. J of Psychophysiology). A recent paper which found, in line with the reported findings, decreased beta power prior to expected visual events (based on prior learning of stimulus sequences) is the one by Bogaerts, Richter, Landau & Frost (2020, J Neuroscience), which might also be interesting to discuss.

We appreciate the reviewer's feedback regarding the direction of beta modulation and their suggestion to consider research on statistical learning. To address this, we have added the following paragraph to the discussion, providing a broader context for our findings within the existing literature.

discussion section:

p. 25, l. 514ff

We observed lower pre-stimulus beta power in trials with high stimulus probability, which were behaviorally linked to a lower detection threshold. While previous studies associate higher pre-stimulus beta power with predictable stimulus onsets (Arnal & Giraud, 2012), our findings cannot be explained by temporal prediction, as stimulus timing was equally predictable across conditions. An alternative interpretation relates to the role of beta oscillations in maintaining cognitive states. Though not designed to investigate working memory, our results align with the idea that higher beta power reflects resistance to change (Engel & Fries, 2010), as trials with higher pre-stimulus beta were associated with a more conservative detection threshold. Another perspective comes from research on statistical learning. Bogaerts et al. found lower pre-stimulus beta power within learned sequences compared to sequence transitions (Bogaerts et al., 2020). However, their study involved sequence learning, whereas pseudo-randomisation of stimuli and probability cues prevented learned predictions in our paradigm. The proposed role of beta power in encoding uncertainty is intriguing, however the current design and analysis do not allow to support this relationship.

#3 Several methodological choices would benefit from additional justification:

- Sample Sizes: While the sample sizes appear reasonably large, but it is not clear how they were determined.

To the best of our knowledge, our study is the first study to investigate experimental manipulations of the decision criterion in somatosensory detection, thus no previous effect size was available. To address the important issue of statistical power, we conducted post-hoc sample size estimations based on the observed effect sizes. The effect sizes have been added to the results section (see below). Using the *statsmodels* package in Python, we calculated post-hoc sample size estimates based on Cohen's d for our main behavioral variable — the criterion change induced by the stimulus probability manipulation. The observed effect sizes indicated that a sample of 18 participants in the stable environment and 13 in the volatile environment was required to detect an effect with 80% power at an alpha level of .05.

Results section:

p. 6, l.134

The effect size was medium to large for the criterion change in both studies (Cohen's $d_{stable} = .72$, Cohens' $d_{volatile} = .84$).

p. 6, l. 144

The effect size for the change in sensitivity was small in both studies (Cohen's $d_{stable} = .18$, Cohen's $d_{volatile} = .07$).

- Response History Bias Definition: The response history bias (also labeled implicit bias) is defined based on the previous choice (n-1). Why not include a longer choice history (e.g., n-2 or n-3)?

We focused on the previous choice only, as previous studies suggested that the previous choice (n-1) has the strongest influence on the subsequent choice (Urai et al., 2017, Fritsche et al., 2024). To maximize statistical power, we included only this predictor in our modelling, as it was expected to have the greatest weight on the detection response.

In response to the reviewer's question regarding further lags, we computed autocorrelation values for up to five lags for the choice variable in both studies. The attached plots show the mean autocorrelation and standard deviation in each study. In the stable environment, choice (detection response) autocorrelation gradually decreases with increasing lags, whereas in the volatile environment, autocorrelation is weaker and more variable across participants. Although it would be intriguing to include additional lags in a follow-up analysis, we avoid doing so in the current study as our GLMM is already quite complex.

- Sensor Selection for Pre-Stimulus Analysis: CP4, which reveals the larger P50, was selected as the sensor of interest for the pre-stimulus time-frequency analysis. Could the authors elaborate on this choice? Wouldn't it also be reasonable to expect divergent spatial localization of pre-stimulus versus evoked brain signatures? More generally, the rationale for selecting a single sensor yet doing source reconstruction was not entirely clear to me.

We agree with the reviewer that stimulus-evoked activity and pre-stimulus activity does not necessarily need to come from the same sources in the brain. We based our initial sensor-level analysis on approaches previously used in the visual domain, showing that criterion changes in visual perception are related to pre-stimulus modulations in sensory areas (Kloosterman et al., 2019; Zhou et al., 2021). Accordingly, to maximize statistical power, we selected the channel with the strongest response to the somatosensory stimulus and chose this channel as our channel that is maximally sensitive to somatosensory stimuli.

However, as the reviewer rightly pointed out, we then wondered whether the observed beta modulation at the sensor level truly originates from somatosensory areas or is instead picked up from a remote source due to EEG volume conduction. To investigate this, we applied a beamforming approach to reconstruct the pre-stimulus beta power modulation across the whole brain. This analysis confirmed that the beta power modulation was indeed maximal in somatosensory areas.

To clarify our analysis approach going from sensor to source level we added the following paragraph to the results section:

p. 12, l. 270ff

The observation of lower pre-stimulus beta power at an electrode maximally sensitive to somatosensory stimulation does not provide direct evidence for a somatosensory source. To localize the origin of this modulation, we applied a beamformer to the sensor level data to reconstruct pre-stimulus beta power at the whole-brain source level.

- Pre-Stimulus Beta Power Window: On page 14, it is mentioned that beta power across the entire pre-stimulus window was averaged. However, Figure 3 shows that the effect is primarily present in the final 300 ms leading up to stimulus onset (-0.3 to 0 s).

Since we did not have a strong hypothesis about when the probability cue would show its maximum effect in the pre-stimulus window, we included the entire time window up to stimulation, excluding only the final 100 ms to prevent potential post-stimulus leakage. To ensure comparability between studies, we used the same time window for the stable environment, where we did not have a probability cue in each trial.

While t-values were indeed strongest in the 300 ms preceding the stimulus, the cluster-based permutation test does not indicate that this is the only significant time window. Rather, it suggests a significant difference across the entire tested interval (Sassenhagen & Draschkow, 2019).

More minor Issues:

- In Figures 3 and 4, the axes are not consistent across different panels. Aligning these axes would facilitate accurate interpretation of the figures.

We thank the reviewer for pointing this out and changed the figure color bar limits in figure 3 and the y limits in figure 4 to make it easier to compare results between studies.

Ai Stable environment

Bi Volatile environment

ii previous yes - no response

ii previous yes - no response

new fig. 3 - same colorbar limits for the same contrasts

Marginal effects: Stable environment

Marginal effects: Volatile environment

new fig. 4: same y limits for A & B

- It is unclear what the "beta estimate above Pre-stimulus beta power" reflects in Figure 6. Providing additional detail would help readers understand this aspect.

We agree that the labelling in figure 6 is imprecise. We added to figure 6 that the beta estimate of the mediator (pre-stimulus beta power) reflects the proportion mediated (Tingley et al., 2014).

new fig. 6, proportion mediated added

Reviewer #2 (Remarks to the Author):

The study explores how expectations and prior actions influence perception, focusing on pre-stimulus brain oscillations as potential markers of these biases. Two EEG experiments examined implicit biases (previous choices) and explicit biases (informed stimulus probabilities) in somatosensory detection tasks under volatile and stable contexts. Behavioral results showed decision criteria and confidence ratings were shaped by stimulus probability cues and choice history. Spectral EEG analysis revealed that pre-stimulus beta power from sensory and higher-order cortical areas predicted explicit and implicit biases, respectively, and mediated the effects of prior choices and probabilities on detection responses. This suggests pre-stimulus beta oscillations reflect neural mechanisms underlying these biases.

The results are interesting and the paper presents a thorough investigation of which factors determine oscillatory brain dynamics in relation to different types of bias. I have two points for a small revision:

We thank the reviewer for the positive evaluation and the time taken to review our work.

- The two experiments are discussed in two different figures, and the similarities and differences between the studies are not so clear. If it would be possible to combine design and results of both studies in one figure that would be helpful to grasp the overall experimental structure.

We agree with the reviewer and now present the model and experimental paradigm in figure 1 and the behavioural results for both studies in figure 2. We hope that this makes it easier for the reader to compare the designs and behavioural results.

Fig. 1: Stimulus probability manipulation in a somatosensory perceptual detection task. **A: Signal detection theory model:** According to SDT, valid information about stimulus probabilities changes the decision criterion c while sensitivity $Dprime$ should not be affected. **B: Stable environment:** Participants were presented with a valid probability cue (low or high) at the beginning of each block. Each block consisted of 12 trials, with the proportion of near-threshold and “catch” (i.e., no stimulus) trials according to the probability cue. **C: Volatile environment:** Participants were presented with a probability cue (orange or blue circle) at the beginning of each trial. Abbreviations: **S:** Signal, **N:** Noise.

Fig. 2: Experimentally controlled stimulus expectations shift detection thresholds in stable and volatile probability environments. A: SDT analysis in the stable (A) and volatile environment (B): Participants had a higher hit rate (i) as well as a higher false alarm rate (ii) in the high expectation condition with no significant difference in Dprime (iii). The criterion c was significantly more conservative in the high probability condition (iv). The mean of high confidence ratings in **correct** trials was higher in trials in which the response matched the participants' expectations

(congruent trials) (v). See also **S1.1, S1.2, S1.3, S1.4**, **Significance:** *** $p < .001$, ** $p < .01$, * $p < .05$. **Abbreviations:** ns = not significant.

- The current study focuses on beta oscillations, whereas many previous studies find a clear link between alpha power and decision bias. It would be interesting to read more about why beta occurs more prominently in somatosensory decisions. Relatedly, both alpha and beta is suppressed during more liberal decision making. What does it mean when an oscillation is reduced - i.e. how can it be a meaningful signal? Ole Jensen et al. have tried to answer this question in several papers and reviews, but it still remains an open question. A short discussion would be helpful for interpreting beta in this paper.

We thank the reviewer for raising this insightful point. We added the following paragraph to the discussion which discusses the role of beta power (not only) in somatosensory perception.

p. 25, l. 514ff

Decreased alpha rhythm has been linked to higher detection rates in both visual (Iemi et al., 2017) and somatosensory detection tasks (Al et al., 2020; Craddock et al., 2017), potentially via enhancing excitability in sensory areas (Linkenkaer-Hansen, 2004; Stephani et al., 2021) through baseline shift mechanisms (Samaha et al., 2020). Notably, lower beta power in somatosensory regions has also been associated with increased detection rates (Haegens et al., 2011; Shin et al., 2017). We propose that pre-stimulus beta power modulations play a dominant role in tasks where top-down influences dominate over spontaneous baseline shifts (Engel et al., 2001; Engel & Fries, 2010; Spitzer & Haegens, 2017). Supporting this idea, beta power modulations related to top-down processing have been observed in regions typically dominated by alpha activity (Limanowski et al., 2020).

In contrast, the role of alpha power in decision criterion manipulations within the visual system remains uncertain, as recent studies have reported conflicting findings (Kloosterman et al., 2019; Zhou et al., 2021). One possible explanation for the varying contributions of alpha and beta power in somatosensory perception is the dynamic reconfiguration of brain networks. Recent work by Sharma and colleagues (Sharma et al., 2023, preprint) suggests a gradual shift from parietal alpha to frontal network engagement in the pre-stimulus window of a somatosensory detection task. This perspective suggests that studies focused on localized brain dynamics may capture transient states within a broader, interconnected network. In this framework, lower pre-stimulus alpha and beta power in sensory areas during trials with a reduced perceptual threshold may reflect efficient information routing within different states of the same network.

- Al, E., Iliopoulos, F., Forschack, N., Nierhaus, T., Grund, M., Motyka, P., Gaebler, M., Nikulin, V. V., & Villringer, A. (2020). Heart–brain interactions shape somatosensory perception and evoked potentials. *Proceedings of the National Academy of Sciences*, *117*(19), 10575–10584. <https://doi.org/10.1073/pnas.1915629117>
- Arnal, L. H., & Giraud, A.-L. (2012). Cortical oscillations and sensory predictions. *Trends in Cognitive Sciences*, *16*(7), 390–398. <https://doi.org/10.1016/j.tics.2012.05.003>
- Bogaerts, L., Richter, C. G., Landau, A. N., & Frost, R. (2020). Beta-Band Activity Is a Signature of Statistical Learning. *The Journal of Neuroscience*, *40*(39), 7523–7530. <https://doi.org/10.1523/JNEUROSCI.0771-20.2020>
- Burnham, K. P., & Anderson, D. R. (Hrsg.). (2004). *Model Selection and Multimodel Inference*. Springer New York. <https://doi.org/10.1007/b97636>.
- Craddock, M., Poliakoff, E., El-dereby, W., Klepousniotou, E., & Lloyd, D. M. (2017). Pre-stimulus alpha oscillations over somatosensory cortex predict tactile misperceptions. *Neuropsychologia*, *96*, 9–18. <https://doi.org/10.1016/j.neuropsychologia.2016.12.030>
- Eckert, A.-L., Gounitski, Y., Guggenmos, M., & Sterzer, P. (2023). Cross-Modality Evidence for Reduced Choice History Biases in Psychosis-Prone Individuals. *Schizophrenia Bulletin*, *49*(2), 397–406. <https://doi.org/10.1093/schbul/sbac168>
- Engel, A. K., & Fries, P. (2010). Beta-band oscillations—Signalling the status quo? *Current Opinion in Neurobiology*, *20*(2), 156–165. <https://doi.org/10.1016/j.conb.2010.02.015>
- Engel, A. K., Fries, P., & Singer, W. (2001). Dynamic predictions: Oscillations and synchrony in top–down processing. *Nature Reviews Neuroscience*, *2*(10), 704–716. <https://doi.org/10.1038/35094565>
- Fritsche, M., Majumdar, A., Strickland, L., Liebana Garcia, S., Bogacz, R., & Lak, A. (2024). Temporal regularities shape perceptual decisions and striatal dopamine signals. *Nature Communications*, *15*(1), 7093. <https://doi.org/10.1038/s41467-024-51393-8>
- Haegens, S., Nácher, V., Hernández, A., Luna, R., Jensen, O., & Romo, R. (2011). Beta oscillations in the monkey sensorimotor network reflect somatosensory decision making. *Proceedings of the National Academy of Sciences*, *108*(26), 10708–10713.

<https://doi.org/10.1073/pnas.1107297108>

- Kloosterman, N. A., de Gee, J. W., Werkle-Bergner, M., Lindenberger, U., Garrett, D. D., & Fahrenfort, J. J. (2019). Humans strategically shift decision bias by flexibly adjusting sensory evidence accumulation. *eLife*, *8*, e37321. <https://doi.org/10.7554/eLife.37321>
- Limanowski, J., Litvak, V., & Friston, K. (2020). Cortical beta oscillations reflect the contextual gating of visual action feedback. *NeuroImage*, *222*, 117267. <https://doi.org/10.1016/j.neuroimage.2020.117267>
- Linkenkaer-Hansen, K. (2004). Prestimulus Oscillations Enhance Psychophysical Performance in Humans. *Journal of Neuroscience*, *24*(45), 10186–10190. <https://doi.org/10.1523/JNEUROSCI.2584-04.2004>
- Samaha, J., Iemi, L., Haegens, S., & Busch, N. A. (2020). Spontaneous Brain Oscillations and Perceptual Decision-Making. *Trends in Cognitive Sciences*, *24*(8), 639–653. <https://doi.org/10.1016/j.tics.2020.05.004>
- Sassenhagen, J., & Draschkow, D. (2019). Cluster-based permutation tests of MEG/EEG data do not establish significance of effect latency or location. *Psychophysiology*, *56*(6), e13335. <https://doi.org/10.1111/psyp.13335>
- Sharma, A., Lange, J., Vidaurre, D., & Florin, E. (2023). *Spontaneous network transitions predict somatosensory perception*. <https://doi.org/10.1101/2023.10.19.563130>
- Shin, H., Law, R., Tsutsui, S., Moore, C. I., & Jones, S. R. (2017). The rate of transient beta frequency events predicts behavior across tasks and species. *eLife*, *6*, e29086. <https://doi.org/10.7554/eLife.29086>
- Spitzer, B., & Haegens, S. (2017). Beyond the Status Quo: A Role for Beta Oscillations in Endogenous Content (Re)Activation. *Eneuro*, *4*(4), ENEURO.0170-17.2017. <https://doi.org/10.1523/ENEURO.0170-17.2017>
- Stephani, T., Hodapp, A., Jamshidi Idaji, M., Villringer, A., & Nikulin, V. V. (2021). Neural excitability and sensory input determine intensity perception with opposing directions in initial cortical responses. *eLife*, *10*, e67838. <https://doi.org/10.7554/eLife.67838>

Tingley, D., Yamamoto, T., Hirose, K., Keele, L., & Imai, K. (2014). **mediation**: R Package for Causal Mediation Analysis. *Journal of Statistical Software*, 59(5).

<https://doi.org/10.18637/jss.v059.i05>

Urai, A. E., Braun, A., & Donner, T. H. (2017). Pupil-linked arousal is driven by decision uncertainty and alters serial choice bias. *Nature Communications*, 8(1), 14637.

<https://doi.org/10.1038/ncomms14637>

Zhou, Y. J., Iemi, L., Schoffelen, J.-M., de Lange, F. P., & Haegens, S. (2021). Alpha Oscillations Shape Sensory Representation and Perceptual Sensitivity. *The Journal of Neuroscience*, 41(46), 9581–9592. [https://doi.org/10.1523/JNEUROSCI.1114-](https://doi.org/10.1523/JNEUROSCI.1114-21.2021)

21.2021